

# Centroids in second-order conservative remapping schemes on spherical coordinates

Fuyuki SAITO[1]

[1]Japan Agency for Marine-Earth Science and Technology (JAMSTEC), Yokohama, Japan

**Correspondence:** Fuyuki SAITO (saitofuyuki@jamstec.go.jp)

**Abstract.** The transformation of data from one grid system to another is common in climate studies. Among the many schemes used for such transformations is second-order conservative remapping. In particular, a second-order conservative remapping scheme first introduced in 1987 and extended in 1999 to work on the general grids of a sphere has, either directly or indirectly, has served as an important base in a variety of studies.

In this study, the author describes a fundamental problem in the derivation of the method proposed by a pioneer study relating to the treatment of the centroid used as a reference point for the second-order terms in the longitudinal direction. In principle, use of the original formulation may cause damage to the entire remapping result. However, a method's native implementation software includes a preprocessing procedure that tends to minimize or even erase the error as a side effect in many, if not most, typical applications. In this study, three alternative formulations are proposed and tested and are shown to work in a simple application.

## 1 Introduction

Numerical climate models commonly couple individual component models such as models for atmosphere, ocean, and land. These component models are typically developed as stand-alone models and often adopt their own grid system for efficiency. Coupling between such components involves field transformations of data from one grid system to another, while preserving key attributes of interest, e.g., global and/or local integrals. This procedure for conservative quantities is often referred to as *conservative remapping*(e.g. Dukowicz and Kodis, 1987). As summarized in Mahadevan et al. (2022), there have been considerable efforts to create conservative remapping algorithms for various problems.

Remapping algorithms used in global climate studies are typically based on first- and second-order conservative mesh-based schemes (Mahadevan et al., 2022). In the first-order conservative scheme, a conservative quantity assuming a constant distribution over the source grid cell is transformed into the overlapped destination grid cells with area-weighted remapping (Bryan et al., 1996). On the other hand, in the second-order conservative scheme, a linear distribution within a source grid cell is assumed, which results in a more accurate and smoother transformation than is the case for first-order schemes. In particular, a second-order algorithm works efficiently when remapping from spatially coarse resolution to fine resolution. Because of this, it is considered the preferred choice in many remapping applications. Dukowicz and Kodis (1987) (hereafter referred to as DK87) first provided a second-order conservative remapping algorithm that works for any general grid system using





Gauss's divergence theorem for simplification of area integrals converted into line integrals. According to Taylor (2024), most conservative remapping algorithms are variants of this approach (there is also a good summary of the remapping method in the appendix of Taylor, 2024).

Jones (1999), hereafter referred to as J99, extends the DK87 theory to spherical coordinates, offering an approach that can be applied to any type of grid on a sphere. Many efforts to maximize efficiency are included in the proposed algorithm, and a number of problems essentially originating from the spherical coordinate system are solved. J99 also provides the Spherical Coordinate Remapping and Interpolation Package (SCRIP), a native software to implement the algorithm (see https://github. com/SCRIP-Project) in addition to four other remapping methods. SCRIP is one of the most widely used remapping software packages in the climate community (Ullrich et al., 2009). For example, Climate Data Operators (CDO) (Schulzweida, 2023) includes a conservative remapping option that incorporates SCRIP with rewriting the source code from Fortran to ANSI/C. In addition, SCRIP has been adopted by the general coupler library OASIS3-MCT_3.0 (Craig et al., 2017), which is used by many modeling groups.

The algorithms and the software proposed in J99 have, either directly or indirectly, been an important base in a variety of studies, including both observational and model data analyses(e.g. Barnes et al., 2024), as well as numerical model development (e.g. Ding et al., 2024). Despite this widespread acceptance, however, there appears to be one distinct and fundamental problem in the derivation of core equations in J99 that, to the author's knowledge, has not previously been recognized nor reported.

The problem is in the treatment of a reference point to evaluate the second-order term in the longitudinal direction. In J99, one of the core equations is, at the very end, transformed into an invalid formulation. If one implements the J99 algorithm following the equations as presented, there is a risk that serious damage will be caused to the remapping result.

Although few, if any, studies using the second-order conservative remapping scheme in SCRIP have reported strange or erroneous behavior, this is not because the derivation is valid. Rather, there is a small preprocessing block in the source code that adjusts some of the key variables for possibly other objectives which can mask the fundamental problem as a side effect. With this adjustment, any errors originating from the invalid derivation tend to be minimized. In fact, the errors can be fully canceled when the source grid cell is a simple one, such as a regular latitude-longitude (RLL) rectangle.

In the next section, the basics of second-order conservative remapping methods are described, with a proposal for a consistent formulation of the scheme. The fundamental problem in the J99 derivation is identified, and the reasons why the invalid derivation has not heretofore been revealed as a problem are discussed. In the third section of the paper, the influence of the inconsistent formulation is demonstrated in simple but practical cases. An experiment showing a sample implementation of the proposed schemes is presented.

## 2 Description of the second-order remapping methods

This section describes the basic idea of the second-order conservative remapping scheme of DK87 and its extension to the spherical coordinate system as formulated by J99. The original equations and terms are transformed into the formulation





shown in J99. For example, the volume integral notation in DK87 is replaced by the surface integral in accordance with J99. Additionally, some new symbols unique to the present paper are introduced for description.

## 2.1 Derivation on a general case

Below is the set of equations derived in a slightly different way, partially following the DK87 method. The derivation is somewhat roundabout but is necessary for the flow of the present paper. Some trivial descriptions are included so as to avoid ambiguities, which the author believes is necessary in the context of the present paper.

The object is to compute in a conservative manner a flux term on a destination grid from the flux term on a source grid over a surface of three-dimensional Euclidean space. For any flux terms that must satisfy a constraint to preserve conservation, the flux integral over each source grid cell must be consistent with the average value in the grid cell as follows:

$$\overline{f}_n A_n = \int_{A_n} f_n \, \mathrm{d}A, \tag{1}$$

where $n$ is the source cell index, $f_n$ and $\overline{f}_n$ are a flux term and its average over the area of source cell $n$, respectively. Equation (1) corresponds to Eq. (19) in DK87. Also, Eq. (1) is identical to Eq. (4) in J99.

DK87 proposes to approximate the source flux by a combination of the average and its gradient. Assuming the flux gradient is constant across a source grid cell locally, the flux can be approximated using Taylor series expansion around a reference point as follows:

$$f_n = f(\mathbf{c}_n) + \nabla_n f \cdot (\mathbf{r} - \mathbf{c}_n), \tag{2}$$

where $\mathbf{c}_n$ is the position vector of a reference point (corresponding to $\overline{\mathbf{r}}_k$ in DK87) and $\nabla_n f$ is a gradient of $f$ in source grid cell $n$. The reference $\mathbf{c}_n$ can be chosen arbitrarily in the source cell; here, it is defined such that the flux approximation Eq. (2) satisfies the condition Eq. (1). By substituting $f_n$, the following condition is obtained:

$$\overline{f}_n A_n = f(\mathbf{c}_n) A_n + \int_{A_n} (\nabla_n f \cdot \mathbf{r}) \, \mathrm{d}A - \int_{A_n} (\nabla_n f \cdot \mathbf{c}_n) \, \mathrm{d}A. \tag{3}$$

In order to satisfy Eq. (3) for any flux and its gradient, the following constraints are obtained:

$$\overline{f}_n = f(\mathbf{c}_n), \tag{4}$$

$$\int_{A_n} (\nabla_n f \cdot \mathbf{r}) \, \mathrm{d}A - \int_{A_n} (\nabla_n f \cdot \mathbf{c}_n) \, \mathrm{d}A = 0. \tag{5}$$

Given a constant gradient across source grid cell $n$, the gradient terms in Eq. (5) can be taken out of the integral, at least over the three-dimensional Cartesian coordinate system, as follows:

$$\nabla_n f \cdot \int_{A_n} \mathbf{r} \, \mathrm{d}A - \nabla_n f \cdot \mathbf{c}_n \int_{A_n} \mathrm{d}A = 0. \tag{6}$$





Thus, the reference $\mathbf{c}_n$ can be inverted as

$$85 \quad \mathbf{c}_n = \int\limits_{A_n} \mathbf{r}\,\mathrm{d}A \bigg/ \int\limits_{A_n} \mathrm{d}A = \frac{1}{A_n} \int\limits_{A_n} \mathbf{r}\,\mathrm{d}A. \tag{7}$$

Under Eq. (7) constraints, with the flux approximation

$$f_n = \overline{f}_n + \nabla_n f \cdot (\mathbf{r} - \mathbf{c}_n), \tag{8}$$

the flux term automatically satisfies the conservation characteristics of Eq. (1). Conservation is preserved with second-order accuracy if the gradient is at least a first-order approximation; if the second term is neglected, the method corresponds to the

90 first-order method.

The position computed in Eq. (7) corresponds to the geometric center, often referred to as the *centroid*, of the source grid cell $n$ under the geometry of the target Euclidean space. In the derivation of DK87, the term centroid is used to label the reference point before the condition in Eq. (6) is presented. While this is viable, the author believes it to be slightly more natural to first describe the condition of the reference point and to subsequently describe its coincidence to the centroid, at least in the context

of the present study. Such a derivation is also given in the next section.

The author speculates that it is non-trivial to satisfy transformation from Eq. (5) to Eq. (6) for general coordinates. Regardless of whether this is true, however, it does not matter to the argument of the present paper, and it is sufficient that in the following discussion, Eq. (7) holds for Cartesian coordinates.

## 2.2 Extension to spherical coordinates

Given the topic of the present paper, it may seem strange to declare the final statement of the previous section and to limit the derivation to Cartesian coordinates. To the contrary, the expansion of the DK87 scheme to spherical coordinates presented by J99 is essentially the same as Cartesian coordinates. It is not the expansion to a three-dimensional spherical coordinate system that is equivalent to three-dimensional Euclidean space (hereafter referred to as *physical* spherical coordinates); rather, it is the flat plane of rectangular coordinates specified by latitude and longitude (referred to as *logical* plane coordinates) with the

inclusion of some characteristics of a spherical surface. In J99, there is no explicit definition of position vector $\mathbf{r}$, even though the derivation is implicitly based on the vector expression $\mathbf{r} = [\theta, \phi]^{\mathsf{T}} = \theta\mathbf{e}_\theta + \phi\mathbf{e}_\phi$ on the logical plane coordinates, where the bases $\mathbf{e}_\theta$ and $\mathbf{e}_\phi$ are compatible with two-dimensional Cartesian coordinates. The symbols $\theta$ and $\phi$ are adopted for the latitude and longitude coordinates, respectively, throughout the present paper.

Using the basic idea of the logical plane coordinates, the derivation in the previous section can be repeated. Based on the

110 author's understanding, the following is the background story of the J99 derivation. The Taylor expansion of the source flux, Eq. (2), can be rewritten as

$$f_n = f(\theta_p, \phi_p) + \left(\frac{\partial f}{\partial \theta}\right)_p (\theta - \theta_p) + \left(\frac{\partial f}{\partial \phi}\right)_p (\phi - \phi_p), \tag{9}$$

which is just the analogue to the $(x, y)$ Cartesian representation. A new symbol is introduced to designate a reference point on the logical plane as $\mathbf{p}_n = [\theta_p, \phi_p]^{\mathsf{T}}$, hereafter symbolically referred to as *pivot*, in order to distinguish it from the centroid



described in the previous section. The flux derivatives are evaluated at the reference point $\mathbf{p}_n$. It is formally possible to provide

the derivative terms by the gradient expressed on the physical coordinate system:

$$f_n = f(\theta_p, \phi_p) + \left(\frac{\partial f}{\partial \theta}\right)_p (\theta - \theta_p) + \left(\frac{1}{\cos\theta}\frac{\partial f}{\partial \phi}\right)_p \cos\theta_p(\phi - \phi_p). \tag{10}$$

A compatible formulation to those presented by J99 can be obtained under the assumption of a constant flux gradient across

the grid cell that may not be defined on the pivot, and substituting the final $\cos\theta_p$ term by $\cos\theta$ in Eq. (10), as follows:

$$f_n = f(\theta_p, \phi_p) + \left(\frac{\partial f}{\partial \theta}\right)_n (\theta - \theta_p) + \left(\frac{1}{\cos\theta}\frac{\partial f}{\partial \phi}\right)_n \cos\theta(\phi - \phi_p). \tag{11}$$

A final adjustment to $\theta$ might be reasonable depending on the characteristics of the derivative in longitude. For example, the

logical relative length in terms of $\cos\theta(\phi - \phi_p)$ could be better for some fields.

The source flux formulation of Eq. (11) is now introduced to the flux integration corresponding to Eq. (1) on the logical

plane. In this case, transformation is necessary from a tiny, infinitesimal area, $\mathrm{d}\theta\,\mathrm{d}\phi$, on the logical plane with a kind of density

function, $\sigma(\theta, \phi)$. For equivalence to the physical coordinates,

$$\int_{\text{physical}} f\,\mathrm{d}A = \int\int f\sigma(\theta,\phi)\,\mathrm{d}\theta\,\mathrm{d}\phi, \tag{12}$$

which shows naturally $\sigma\,\mathrm{d}\theta\,\mathrm{d}\phi$ is identical to the area element of the physical spherical coordinates. Therefore, the same

expression of $\mathrm{d}A$ can also be used on the logical plane without risking confusion.

Substituting Eq. (11) into Eq. (1), and using the same area element representation, the conservation condition corresponding

to Eq. (3) is formulated as

$$\overline{f}_n A_n = f(\mathbf{p}_n) A_n + \int_{A_n} \left[\left(\frac{\partial f}{\partial \theta}\right)_n (\theta - \theta_p) + \left(\frac{1}{\cos\theta}\frac{\partial f}{\partial \phi}\right)_n (\phi - \phi_p)\cos\theta\right]\mathrm{d}A. \tag{13}$$

The pivot $\mathbf{p}_n$ reference point can be defined such that Eq. (13) is satisfied for any flux and its derivatives as follows:

$$\overline{f}_n = f(\mathbf{p}_n), \tag{14}$$

$$\int_{A_n} (\theta - \theta_p)\,\mathrm{d}A = 0, \tag{15}$$

$$\int_{A_n} [\cos\theta(\phi - \phi_p)]\,\mathrm{d}A = 0, \tag{16}$$

where the flux derivatives are assumed to be constant across the source grid cell $n$. Equations (15) and (16) work as the

constraints for the pivot coordinates $(\theta_p, \phi_p)$ in the logical plane coordinate system, which can be inverted as

$$\theta_p = \left(\int_{A_n} \theta\,\mathrm{d}A\right) \Big/ \left(\int_{A_n} \mathrm{d}A\right), \tag{17}$$

$$\phi_p = \left(\int_{A_n} \phi\cos\theta\,\mathrm{d}A\right) \Big/ \left(\int_{A_n} \cos\theta\,\mathrm{d}A\right). \tag{18}$$





This can serve as the formal formulation of the pivot coordinates on the logical plane that the author proposed as the basis of the second-order conservative remapping scheme by J99 when the source flux term is approximated by the expression in Eq. (5).

As previously mentioned, the derivation of the reference point on the logical plane is equivalent to that on the Cartesian plane. Therefore, the centroid condition of Eq. (7) can be incorporated into the logical plane as is. The centroid coordinates

$\mathbf{c}_n = [\theta_c, \phi_c]^\mathsf{T}$ can then be similarly formulated as

$$\theta_c = \left( \int_{A_n} \theta \, \mathrm{d}A \right) \Big/ \left( \int_{A_n} \mathrm{d}A \right) = \frac{1}{A_n} \int_{A_n} \theta \, \mathrm{d}A, \tag{19}$$

$$\phi_c = \left( \int_{A_n} \phi \, \mathrm{d}A \right) \Big/ \left( \int_{A_n} \mathrm{d}A \right) = \frac{1}{A_n} \int_{A_n} \phi \, \mathrm{d}A, \tag{20}$$

respectively, which is accompanied by the flux approximation

$$f_n = f(\theta_c, \phi_c) + \left( \frac{\partial f}{\partial \theta} \right)_n (\theta - \theta_c) + \left( \frac{\partial f}{\partial \phi} \right)_n (\phi - \phi_c). \tag{21}$$

It is clear that the centroid and pivot on the logical plane are identical in the latitudinal direction but different in the longitudinal direction. The denominator of the pivot condition Eq. (18) includes $\cos\theta$ in the integral, which reflects the adjustment of Eq. (11). It is not about which formulation is better or worse. It is possible to reasonably construct an algorithm so that the conservation property holds, whether the former or the latter is adopted.

### 2.3  Inconsistent formulation relating to the centroid

The formulation of J99 is mainly derived for the area-averaged flux over the destination grid cell (after remapping), and is essentially the same as that over the source grid cell. The flux over the destination grid is formulated as follows:

$$\overline{F}_k = \frac{1}{A_k} \sum_{n=1}^{N} \int_{A_{nk}} f_n \, \mathrm{d}A, \tag{22}$$

where $\overline{F}_k$ is the average flux over the destination grid cell $k$, and $A_{nk}$ is the area of the source grid cell $n$ covered by the destination grid cell $k$. The summation is performed for all overlapped cells of $N$. The average flux term at the destination grid

cell can be approximated as follows:

$$\overline{F}_k = \sum_{n=1}^{N} \left[ \overline{f}_n w_{1\,nk} + \left( \frac{\partial f}{\partial \theta} \right)_n w_{2\,nk} + \left( \frac{1}{\cos\theta} \frac{\partial f}{\partial \phi} \right)_n w_{3\,nk} \right], \tag{23}$$





which corresponds to Eq. (7) of J99. The three coefficients, $w_{1nk}$, $w_{2nk}$, $w_{3nk}$, are called the remapping weights and are derived according to J99 as follows:

$$w_{1nk} = \frac{1}{A_k} \int\limits_{A_{nk}} \mathrm{d}A, \tag{24}$$

$$w_{2nk} = \frac{1}{A_k} \int\limits_{A_{nk}} (\theta - \theta_n)\, \mathrm{d}A, \tag{25}$$

$$w_{3nk} = \frac{1}{A_k} \int\limits_{A_{nk}} \cos\theta(\phi - \phi_n)\, \mathrm{d}A. \tag{26}$$

Note that Eqs. (25) and (26) are presented as intermediate formulations during the derivation. The reference point $\theta_n, \phi_n$ is actually called the centroid in J99. From Eqs. (23) and (26), however, it can be deduced that the reference is equivalent to the formulation of Eq. (11) rather than Eq. (10), since the $\cos\theta$ term appears in the integral.

The final formulations of $w_{2nk}$ and $w_{3nk}$ are as follows:

$$\underset{\mathrm{ORG}}{w_{2nk}} = \frac{1}{A_k} \int\limits_{A_{nk}} \theta\, \mathrm{d}A - \frac{w_{1nk}}{A_n} \int\limits_{A_n} \theta\, \mathrm{d}A, \tag{27}$$

$$\underset{\mathrm{ORG}}{w_{3nk}} = \frac{1}{A_k} \int\limits_{A_{nk}} \phi\cos\theta\, \mathrm{d}A - \frac{w_{1nk}}{A_n} \int\limits_{A_n} \phi\cos\theta\, \mathrm{d}A, \tag{28}$$

which correspond to Eqs. (9) and (10) in J99, respectively. Based on the author's exploration of the SCRIP source code and the CDO source code, the formulations correspond to Eqs. (27) and (28) being implemented. Substituting the pivot condition Eq. (17), the remapping weight in the latitudinal direction, $w_{2nk}$ of Eq. (25), is transformed into the formulation using the pivot term $\mathbf{p}_n = [\theta_p, \phi_p]^\mathsf{T}$ as

$$w_{2nk} = \frac{1}{A_k} \int\limits_{A_{nk}} [\theta - \theta_p]\, \mathrm{d}A = \frac{1}{A_k} \int\limits_{A_{nk}} \theta\, \mathrm{d}A - \frac{A_{nk}}{A_k}\frac{1}{A_n} \int\limits_{A_n} \theta\, \mathrm{d}A \tag{29}$$

$$\equiv \underset{\mathrm{ORG}}{w_{2nk}},$$

which is equivalent to the original derivation of Eq. (27).

However, the remapping weight for the longitudinal direction determined in Eq. (28) is invalid. Employing a similar procedure to that described above, with the substitution of Eq. (18) into the remapping weight expression for the longitudinal





direction, $w_{3nk}$ can be formulated using the pivot term as follows:

$$w_{3nk} = \frac{1}{A_k} \int_{A_{nk}} [\phi - \phi_p] \cos\theta \, \mathrm{d}A \tag{30}$$

$$= \frac{1}{A_k} \int_{A_{nk}} \phi \cos\theta \, \mathrm{d}A \tag{31}$$

$$- \frac{1}{A_k} \left[ \int_{A_{nk}} \cos\theta \, \mathrm{d}A \bigg/ \int_{A_n} \cos\theta \, \mathrm{d}A \right] \int_{A_n} \phi \cos\theta \, \mathrm{d}A$$

$$\underset{\mathrm{ORG}}{\not\equiv} w_{3nk},$$

where the final expression is not equivalent to Eq. (28). It seems that the two integrals of the $\cos\theta$ terms that appear in brackets in Eq. (31) have been extracted, or that the centroid formulation has superseded somewhere during the derivation.

Consequently, implementing the J99 algorithm following the precise formulation as originally presented may, in fact, damage the remapping result.

How the remapping weights are influenced by the invalid formulation can be demonstrated by using a simple configuration in which both source and destination grids are set as RLL grids on a unit sphere, and the cells are equally spaced along the longitude and latitude. The latitudes and longitudes of the grid lines (cell corner coordinates) are expressed as $\theta = 180°(j - N_\theta/2)/N_\theta, j = 0, \cdots, N_\theta$; and $\phi = 360°(i - 0.5 - N_\phi/2)/N_\phi, i = 0, \cdots, N_\phi$, respectively. The source and destination grids adopt $N_\theta, N_\phi = 64, 128$ and $N_\theta, N_\phi = 128, 256$, respectively, where a source cell contains $2 \times 2 = 4$ destination cells, and a destination cell does not extend over multiple source cells. Figure 1 shows the distribution of the remapping weight $w_{3nk}$ over the example source/destination configuration. One source cell has four remapping weights for each overlapped destination cell; those for the north-west designation cells are plotted in the figure. (It is for this reason that the figure is not symmetric about the equator.) Since the relative orientation of a source cell and its overlapped destination cells is equivalent along the longitudinal direction, the remapping weight must be axisymmetric. Figure 1(a) displays the results for weights computed with the original formulation Eq. (28), clearly showing the breaking of symmetry. In contrast, in Fig 1(b), the remapping weights were computed using the formulation satisfying the pivot condition (Eq. 31), which produces the axisymmetric results shown in the figure. This may be extremely surprising to anyone who uses SCRIP or related tools, as will be discussed in the next section.

## 2.4 What happens in the native implementation

As shown in Fig. 1, the remapping weights computed precisely according to the original formulation (Eq. 28) result in very strange image, one that is actually too strange to survive in any practical application. Indeed, the author doubts that it would pass the review of developers and users since even a very simple benchmark test would result in an unreasonable remapping.

The numerical library SCRIP implements the inconsistent formulation of the second-order conservative remapping scheme as Eqs. (27) and (28). CDO (Schulzweida, 2023) includes a conservative remapping option based on a mostly faithful transport of the SCRIP Fortran source code to ANSI/C. As far as the author can determine, there is no part of the source code that com-





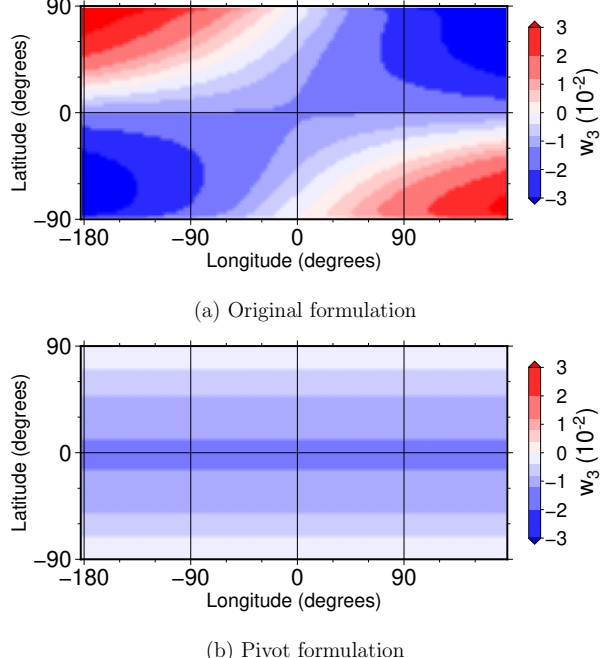

(a) Original formulation

(b) Pivot formulation

**Figure 1.** Demonstration of the remapping weight computation. (a) $w_{3\,nk}$ using the J99 original algorithm (Eq. 28) (b) $w_{3\,nk}$ by Eq. (31).

putes the remapping weights in a way that differs from the original (erroneous) formulation. Why, then, have there apparently been no reports of unreasonable results produced by SCRIP? The answer appears to be that there is a key procedure in the SCRIP source code that eliminates the error originating from the invalid derivation, a procedure which the author suspects is
215 implemented for another reason.

  It is reasonable to conclude that Eq. (31) holds for any longitudinal origin; otherwise, the remapping weight $w_{3\,nk}$ would change its value according to the coordinate (this is exactly how the author discovered the fundamental problem). Thus, in the computation of the weights for each source cell $n$, it would be safe to rotate around the pole by $\phi_{\text{rep}}$, which would simply correspond to replacing the longitudinal variable with a relative one. Put formally, Eq. (30) is reformulated into

$$220 \quad w_{3nk} = \frac{1}{A_k} \int\limits_{A_{nk}} \cos\theta[(\phi - \phi_{\text{rep}}) - (\phi_p - \phi_{\text{rep}})]\,\mathrm{d}A \tag{32}$$

$$= \frac{1}{A_k} \int\limits_{A_{nk}} \cos\theta(\phi - \phi_{\text{rep}})\,\mathrm{d}A - \frac{1}{A_k} \int\limits_{A_{nk}} \cos\theta(\phi_p - \phi_{\text{rep}})\,\mathrm{d}A, \tag{33}$$

which is an identity for any $\phi_{\text{rep}}$. Normally, the pivot longitude $\phi_p$ is an internally computed quantity using Eq. (18). If one can compute $\phi_p$ externally in advance and compute the remapping weights using the longitude relative to $\phi_{\text{rep}} = \phi_p$, then the identical solution can be obtained without changing the formulation of Eq. (30), replacing $\phi$ with $\phi - \phi_p$ and $\phi_p$ with
225 $\phi_p - \phi_p = 0$. Thus, over such a locally rotated coordinate system, the pivot value, the second term in Eq. (30), contributes nothing since it equals zero.





```
phi1 = in_phi1 - grid1_lon
if (phi1 >  pi) then
phi1 = phi1 - pi2
else if (phi1 < -pi) then
phi1 = phi1 + pi2
endif
phi2 = in_phi2 - grid1_lon
if (phi2 >  pi) then
phi2 = phi2 - pi2
else if (phi2 < -pi) then
phi2 = phi2 + pi2
endif
```

**Figure 2.** A part of source code cited from SCRIP. This part is extracted from subroutine `line_integral_phi` in the source file `SCRIP/source/remap_conserv.f`. The numbers on the far left are not part of the source code. This subroutine implements the line integration along a path around a source and destination grid cell, which is transformed into the area integral of Eqs. (24), (27), and (28) using Gauss's divergence theorem. This block operates as a preprocessing step before the line integration. Please see the explanation in the main text.

When the pivot longitude $\phi_p = 0$, the following relation is obtained from Eq. (18):

$$\int_{A_n} \phi \cos\theta \, \mathrm{d}A = 0, \tag{34}$$

over the rotated system. This integral term appears in the inconsistent formulation of the second term in Eq. (28). Although the
230 second term is inconsistent overall, the integral term is computed in a way that is identical to the consistent formulation; only the coefficient makes the term inconsistent. Thus, if the longitude is rotated by $\phi_{\mathrm{rep}} = \phi_p$ in the computation of Eq. (28), Eq. (31), and Eq. (28) provide identical solutions, even though the coefficients are invalid, since these coefficients are essentially erased by the zero-valued integration term. The problem is, of course, that it does not make sense to expect a valid $\phi_p$ based on an inconsistent computation of the remapping weights.
However, there is, indeed, an explanation.

In the SCRIP source code, the procedure to transform the input longitude into a relative one definitely exists, as shown in Fig. 2. In the source code, `in_phi1` and `in_phi2` are dummy input arguments to hold the longitude of two points on a side of source grid cell, which act as start and end points of the line integral. Another dummy input argument `grid1_lon`, hereafter referred to as the *representative* coordinate for the input cell, is utilized as the reference point to compute their relative
longitudes `phi1` and `phi2`. This term works in exactly the same way as $\phi_{\mathrm{rep}}$ that was introduced above. After conversion to the relative longitudes, `phi1` and `phi2` are conditionally adjusted with a constant `pi` as $\pi$ for monotonicity; `phi1` and `phi2`





are then applied for the evaluation of line integration. Note that this adjustment is executed only when the absolute relative longitude is larger than $pi$. According to the comment in the source code, this adjustment seems intended to address the periodic boundary in the longitudinal direction and to specify the internal direction of the cell when the difference in longitude is larger than $180°$ (this may happen around the pole for general grid systems).

What quantities are expected for the representative coordinates $\phi_{\text{rep}}$ are not well defined in the source code, although the actual arguments are named as `grid1_center_lon` or similar, and a comment in the source code indicates that they represent the grid center. The library appears to require the central longitude for each grid cell as a mandatory input field; however, the property of being the center of the cell does not appear to be used. Since only the difference between the two relative longitudes adjusted by the representative longitude is used in the computation, the central value is of no particular significance. It is even possible to have the representative longitude fall outside the cell boundaries since it serves only as reference[1]. Moreover, the definition of central longitude is ambiguous for general shapes of the grid cell. In principle, the reference longitudes are parameters that must be supplied by the user according to the source grid cell configuration and not computed by the library during the computation of the remapping weights nor the actual remapping. Thus, the author supposes that the reference point works solely as a representative.

Whether or not it was intended, the adjustment resulting from the implementation of this part succeeds in resolving all the problems relating to the pivot longitude, since, as a side effect, the adjustment corresponds precisely to the rotation of the coordinates, as explained above.

Although not forced, it is quite natural to set the representative longitude $\phi_{\text{rep}}$ as the center of the longitude range of the source grid cells. One reason for this is that a sample program included in SCRIP makes the computation this way; another is that the center longitude is often used for other situations, e.g., visualization, and thus they can be easily prepared. For some special cases, such as benchmark tests, the central longitude is used to evaluate the flux gradient, which is not generally possible for practical applications. For the RLL rectangle grid cells in spherical coordinates, the center longitude is identical to the pivot longitude, and therefore the rotation helps to cancel the contribution of the pivot term. Moreover, if a cell is symmetric along a meridian, then, naturally, the pivot coordinate coincides with the center longitude. In most cases using various shapes of grid cells, the center longitude defined by the user for particular target grid cells may not be far from the pivot longitude, and the problem of the incorrect contribution of the pivot term can be rendered insignificant, as shown in the previous section. Again, the representative longitude is left to the user's discretion, and these side effects are unexpected.

According to the CDO reference manual, the second-order conservative remapping command (`REMAPCON2`) is not available for unstructured source grids. This is good news for CDO users. Thanks to the mid-longitude representative, there may be no risk that the user will suffer from the inconsistent formulation of the remapping weights. However, the author is not fully convinced, and such a conclusion should be confirmed by an expert in the area.

---

[1]The function of the representative longitude to specify which is the internal seems to be only effective when the cell span is larger than $180°$; thus, it does not matter, at least for RLL rectangle cell cases





### 2.5 Proposal of alternate formulations

Several remedies are possible for the inconsistency problem cited above. In this subsection, three slightly different formulations

are proposed as alternatives to the J99 core equations. The first-order remapping weight, $w_{1\,nk}$, and the second-order remapping weight in the latitudinal direction, $w_{2\,nk}$, are identical to those originally presented in J99 but are listed here for completeness.

The proposed modification is introduced into SCRIP version 1.5, which is hereafter referred to as SCRIP-p to distinguish it from the official SCRIP. The longitude adjustment with the cell representative point shown in Fig. 2 is left as is, since the function of longitude adjustment relative to the representative is, in any event, necessary in order to deal with the periodic

boundary condition in a simple way. The original variable names are left as they were even if they contain the word `centroid` as a misconception, in order to minimize the modification.

The original implementation is identified as Scheme O for reference. This scheme is expected to provide a valid solution when a representative longitude matches the pivot longitude of a source cell, and thus the pivot term contributes virtually nothing to the remapping weights.

#### 2.5.1 Scheme P — *pivot* method

Scheme P is a mostly straightforward implementation of the original J99 formulation, where only the invalid computation of the remapping weight $w_{3\,nk}$ is replaced according to the pivot condition. Formally, the centroid definition must be excluded from the beginning of the implementation as it is incompatible with this formulation. The flux approximation is formulated as

$$\overline{F}_k = \sum_{n=1}^{N} \left[ \overline{f}_n w_{1\,nk\mathsf{P}} + \left( \frac{\partial f}{\partial \theta} \right)_n w_{2\,nk\mathsf{P}} + \left( \frac{1}{\cos\theta} \frac{\partial f}{\partial \phi} \right)_n w_{3\,nk\mathsf{P}} \right], \tag{35}$$

and the corresponding remapping weights are formulated as

$$\begin{cases} w_{1\,nk\mathsf{P}} = \dfrac{1}{A_k} \displaystyle\int\limits_{A_{nk}} \mathrm{d}A, \\[2ex] w_{2\,nk\mathsf{P}} = \dfrac{1}{A_k} \displaystyle\int\limits_{A_{nk}} (\theta - \theta_p)\,\mathrm{d}A = \dfrac{1}{A_k} \displaystyle\int\limits_{A_{nk}} \theta\,\mathrm{d}A - \dfrac{w_{1\,nk}}{A_n} \displaystyle\int\limits_{A_n} \theta\,\mathrm{d}A, \\[2ex] w_{3\,nk\mathsf{P}} = \dfrac{1}{A_k} \displaystyle\int\limits_{A_{nk}} (\phi - \phi_p)\cos\theta\,\mathrm{d}A = \dfrac{1}{A_k} \displaystyle\int\limits_{A_{nk}} \phi\cos\theta\,\mathrm{d}A - \dfrac{1}{A_k} \dfrac{\Omega_{3\,nk}}{\Omega_{3\,n}} \displaystyle\int\limits_{A_n} \phi\cos\theta\,\mathrm{d}A, \\[2ex] \Omega_{3\,nk} = \displaystyle\int\limits_{A_{nk}} \cos\theta\,\mathrm{d}A. \end{cases} \tag{36}$$

The new term $\Omega_{3\,nk}$ to be applied in the evaluation of $w_{3\,nk}$ is introduced. This term is not a remapping weight but is computed with the same procedure as the other three remapping weights. The integral part of $\Omega_{3\,nk}$ is computed by transforming it into a line integral using Gauss's divergent theorem following the J99 method for the other integrals, and is formulated as

$$\int\limits_{A_{nk}} \cos\theta\,\mathrm{d}A = \oint\limits_{C_{nk}} -\frac{\sin\theta\cos\theta + \theta}{2}\,\mathrm{d}\phi. \tag{37}$$





Although the replacement involves only the computation of a single variable, the source code modification would be the most substantial among the three proposals since the treatment of additional variable $\Omega_{3\,nk}$ must be introduced concurrently with the three standard weights.

This is the only compatible formulation among the three to provide the identical solution to that produced by Scheme O
when the representative longitudes match the mid-longitude of the source cells.

### 2.5.2   Scheme Cd — *centroid-derivative* method

Scheme Cd is the implementation that deviates most from the original J99 formulation, whereby the modification is introduced after the definition of the centroid. Formally, maintaining the centroid formulation, the flux approximation must be reformulated.

The flux approximation is replaced with the formulation

$$
\overline{F}_k = \sum_{n=1}^{N} \left[ \overline{f}_n w_{1\,nk\mathsf{Cd}} + \left( \frac{\partial f}{\partial \theta} \right)_n w_{2\,nk\mathsf{Cd}} + \left( \frac{\partial f}{\partial \phi} \right)_n w_{3\,nk\mathsf{Cd}} \right], \tag{38}
$$

and the corresponding remapping weights are formulated as

$$
\begin{cases}
w_{1\,nk\mathsf{Cd}} = \dfrac{1}{A_k} \displaystyle\int_{A_{nk}} \mathrm{d}A, \\[2em]
w_{2\,nk\mathsf{Cd}} = \dfrac{1}{A_k} \displaystyle\int_{A_{nk}} (\theta - \theta_c)\,\mathrm{d}A = \dfrac{1}{A_k} \displaystyle\int_{A_{nk}} \theta\,\mathrm{d}A - \dfrac{w_{1\,nk}}{A_n} \displaystyle\int_{A_n} \theta\,\mathrm{d}A, \\[2em]
w_{3\,nk\mathsf{Cd}} = \dfrac{1}{A_k} \displaystyle\int_{A_{nk}} (\phi - \phi_c)\,\mathrm{d}A = \dfrac{1}{A_k} \displaystyle\int_{A_{nk}} \phi\,\mathrm{d}A - \dfrac{w_{1\,nk}}{A_n} \displaystyle\int_{A_n} \phi\,\mathrm{d}A.
\end{cases} \tag{39}
$$

This method involves not only a modification of the remapping weight computation, but requires that the caller replace the
gradient (including the cosine latitude factor) of the derivative. It is posited that this method is the easiest to extend to a higher-order conservative remapping scheme since the formulation should be mostly compatible with the Cartesian formulation.

The integral part of $w_{3\,nk}$ is computed using Gauss's divergent theorem as follows:

$$
\int_{A_{nk}} \phi\,\mathrm{d}A = \oint_{C_{nk}} -\phi\sin\theta\,\mathrm{d}\phi. \tag{40}
$$

### 2.5.3   Scheme Cg — *centroid-gradient* method

Scheme Cg is a minor variation of above scheme. The centroid condition is introduced, but the gradient term is accepted as input rather than the derivative. The cosine latitude adjustment is absorbed in the remapping weights computation. The flux approximation is formally expressed as

$$
\overline{F}_k = \sum_{n=1}^{N} \left[ \overline{f}_n w_{1\,nk\mathsf{Cg}} + \left( \frac{\partial f}{\partial \theta} \right)_n w_{2\,nk\mathsf{Cg}} + \left( \frac{1}{\cos\theta} \frac{\partial f}{\partial \phi} \right)_n w_{3\,nk\mathsf{Cg}} \right], \tag{41}
$$





and the corresponding remapping weights are formulated as

$$
\begin{cases}
w_{1\,nk\texttt{Cg}} = \dfrac{1}{A_k} \displaystyle\int\limits_{A_{nk}} \mathrm{d}A, \\[2ex]
w_{2\,nk\texttt{Cg}} = \dfrac{1}{A_k} \displaystyle\int\limits_{A_{nk}} (\theta - \theta_c)\,\mathrm{d}A = \dfrac{1}{A_k} \displaystyle\int\limits_{A_{nk}} \theta\,\mathrm{d}A - \dfrac{w_{1\,nk}}{A_n} \displaystyle\int\limits_{A_n} \theta\,\mathrm{d}A, \\[2ex]
w_{3\,nk\texttt{Cg}} = \cos\theta_c \dfrac{1}{A_k} \displaystyle\int\limits_{A_{nk}} (\phi - \phi_c)\,\mathrm{d}A = \cos\theta_c \left[ \dfrac{1}{A_k} \displaystyle\int\limits_{A_{nk}} \phi\,\mathrm{d}A - \dfrac{w_{1\,nk}}{A_n} \displaystyle\int\limits_{A_n} \phi\,\mathrm{d}A \right], \\[2ex]
\theta_c = \dfrac{1}{A_n} \displaystyle\int\limits_{A_n} \theta\,\mathrm{d}A.
\end{cases}
\tag{42}
$$

The centroid latitude is inserted into the remapping weight $w_{3\,nk}$ computation, which has already been computed in the $w_{2\,nk}$ evaluation.

## 3 Experiment and discussion

### 3.1 Configuration of experiments

In the present study, only the domain of the RLL grid on a unit sphere of $N_\theta$ latitudes and $N_\phi$ longitudes, both for the source and destination grids, is examined. The latitudes and longitudes of the grid lines (cell corner coordinates) are expressed as $\theta = 180°(j - N_\theta/2), j = 0, \cdots, N_\theta$; and $\phi = (i/N_\phi), i = 0, \cdots, N_\phi$, respectively (this is slightly different from the domain definition used for the demonstration in Fig. 1, which does not influence the discussion). The size of the source grid cell is set as $(N_\theta, N_\phi) = (64, 128)$. Several destination grid sizes are examined, including $(N_\theta, N_\phi) = (128, 256), (256, 512), (512, 1024)$. A source grid cell is divided equally into some power of 2 of the destination grid cells, and one destination grid cell overlaps with only one source grid cell.

Two idealized experiments were conducted following J99. These also appear in Lauritzen and Nair (2008); Ullrich et al. (2009). A relatively smooth function resembling a spherical harmonic of order 2 and azimuthal wavenumber 2 (named as $Y_2^2$),

$$
\psi = 2 + \cos^2\theta \cos(2\phi),
\tag{43}
$$

and a relatively high-frequency wave similar to a spherical harmonic of order 32 and azimuthal wavenumber 16 (named as $Y_{32}^{16}$),

$$
\psi = 2 + \sin^{16}(2\theta) \cos(16\phi),
\tag{44}
$$

are used as input for the source grid in each experiment. The mid-longitude and mid-latitude coordinates for each cell are used as a reference point to compute $\psi$ and its gradient to input.





The maximum value of the gradient of $Y_{32}^{16}$ is approximately 10-fold larger than that of $Y_2^2$. Since the alternate formulations proposed in the present study are only to replace the remapping weights in the longitudinal direction, the impact of these alternatives depends on the magnitude of the gradients; thus, $Y_{32}^{16}$ is expected to be more sensitive to the fundamental problem than $Y_2^2$.

The performance of the conservative remapping algorithms was evaluated using two to five measures in the previous studies. In the present paper, the second-order norm $l_2$, as well as the infinity norm $l_\infty$, are introduced. Using $\Psi$ as the field after remapping on the destination grid cell, these norms are computed as

$$l_2 = \sqrt{\frac{I\left[\left(\overline{\Psi}_{\text{num}} - \overline{\Psi}_{\text{exact}}\right)^2\right]}{I\left[\left(\overline{\Psi}_{\text{exact}}\right)^2\right]}}, \tag{45}$$

$$l_\infty = \frac{\max\left[\left|\overline{\Psi}_{\text{num}} - \overline{\Psi}_{\text{exact}}\right|\right]}{\max\left[\left|\overline{\Psi}_{\text{exact}}\right|\right]}, \tag{46}$$

where $I$ is the global integral over the destination grid cell

$$I\left[\overline{F}\right] = \sum_k \overline{F}_k A_k. \tag{47}$$

Following Lauritzen and Nair (2008); Ullrich et al. (2009), the numerical generated solution of the term,

$$\overline{\Psi}_k = \frac{1}{A_k} \int\limits_{A_k} \psi \, dA, \tag{48}$$

is applied for the "exact" term $\overline{\Psi}_{\text{exact}}$, which is computed using fourth-order Gaussian quadrature.

### 355   3.2   Influence of the invalid formulation of the pivot term

As shown in 2.4, it is speculated that for RLL rectangular cell cases, the representative longitude virtually works as the pivot longitude in the official SCRIP, which would erase the fundamental problem. For general shapes of grid cells, the representative longitude may not be the same as the pivot longitude. To investigate the sensitivity of this deviation, a simple experiment is presented using the official SCRIP.

In the sensitivity experiments, the representative longitude for a source cell is set as

$$\phi_{\text{rep}} = \frac{\phi_0 + \phi_1}{2} + \alpha(\phi_1 - \phi_0), \tag{49}$$

where $\phi_0$, $\phi_1$ are the longitude boundaries of the source cell and $\alpha$ is a parameter. In the control case, the representative longitude is set as the mid-longitude of each source grid cell ($\alpha = 0$), which is the same as the test program included in the official package. For the sensitivity experiment, eight $\alpha$ values from $0$ to $0.5$ were chosen. The parameter $\alpha = 0.5$ corresponds

to the case where the representative longitude matches the boundary, $\phi_{\text{rep}} = \phi_1$. All tests keep the representative longitude within the cell, which is naturally expected to work as representative (again, the user has full control of this value and thus there is no guarantee of this).



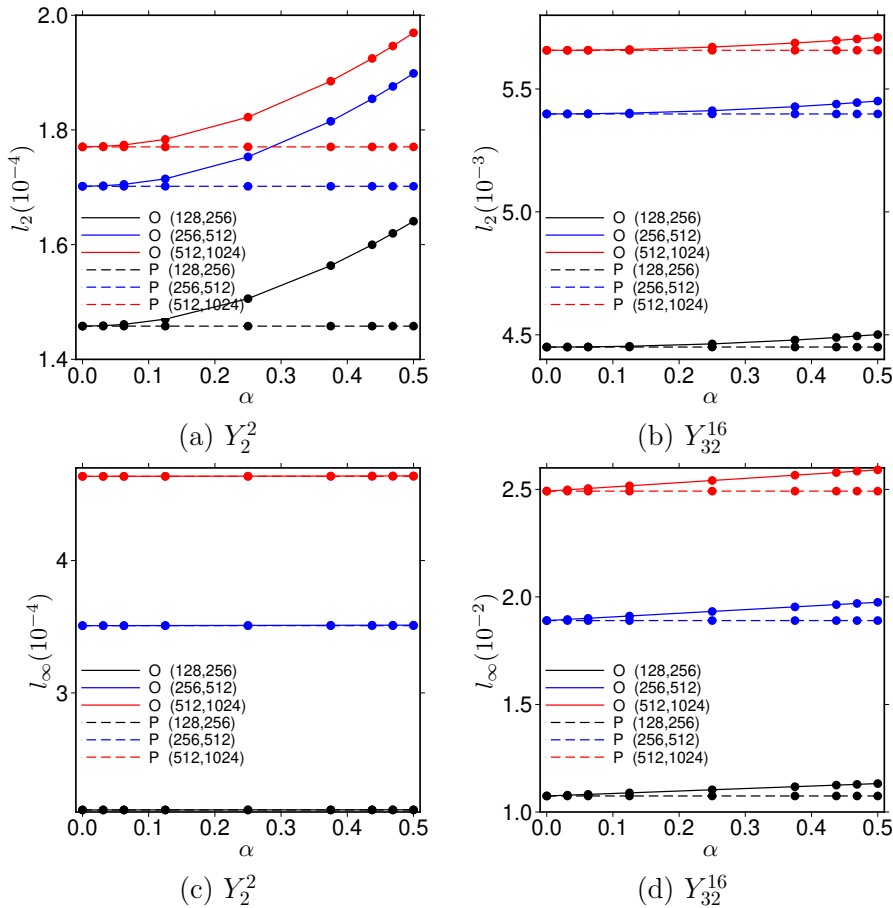

**Figure 3.** Results of shifting the representative longitude from the pivot (mid) longitude within a source cell. The metrics $l_2$ (a,b) and $l_\infty$ (c,d) for the case of three destination grid cells, $(N_\theta, N_\phi) = (128, 256), (256, 512), (512, 1024)$, are plotted. Results of two input fields $Y_2^2$ (a,c) and $Y_{32}^{16}$ (b,d) are shown. Solid and dashed lines correspond to the results for schemes O and P, respectively. Results for schemes O and P nearly overlap in (c).

Since the pivot longitude matches the mid-longitude of the RLL rectangle cell, the influence of a deviation in the pivot longitude can be evaluated in the experiment. Figure 3 shows the results of the sensitivity experiments in terms of the metric $l_2$

versus the representative longitude position within the source cells (solid lines, Scheme O). As shown in the figure, the metric $l_2$ increases as the representative coordinate moves farther from the pivot for all the experiments in the present study. The deviations show similar curves among the destination grids. Even when the representative longitude matches the cell boundary ($\alpha = 0.5$), the metric $l_2$ maintains its order of magnitude. The maximum deviation for $\alpha = 0.5$ is approximately $2.0 \times 10^{-5}$ and $5.0 \times 10^{-5}$ for test $Y_2^2$ and $Y_{32}^{16}$, respectively. Although the user has full control of the representative longitude to be specified,

it is quite natural to expect that it will be within a cell. Thus, practically, even when the representative longitude does not match the pivot, the field after remapping may remain similar without a significant impact.



The metric $l_\infty$ is shown in the lower two panels of Fig. 3. This measure identifies the maximum relative cellwise error over the entire field (Ullrich et al., 2009). In test $Y_2^2$, the impact of the shift in the representative longitudes is kept to approximately $3.9 \times 10^{-7}$; in test $Y_{32}^{16}$, it is approximately $1.0 \times 10^{-3}$. For both metrics, the inconsistency between the representative and pivot

longitudes has less impact than a change in magnitude. For general shapes of the source grid cell, the representative longitude specified by the user may not match the pivot, but even then, it is natural to expect that it will be within the cell. Details of the sensitivities depend on the shape and the magnitude of the flux gradient input, but the impact of the inconsistency should remain small.

An extreme case was also examined in which the representative longitude is set as $\phi_{\mathsf{rep}} \equiv 180°$ for all the source grid cells[2].

The maximum impact of the representative/pivot discrepancies can be determined from this extreme case. Table 1 and Table 2 summarize the metric $l_2$ and $l_\infty$, respectively, for two cases: in one, the representative longitude is set as the mid-longitude of the source cell, the same as in the control experiment above ($\alpha = 0$); in the other, the extreme representative longitude is shared among all the source cells. Tables 1 and 2 show clearly that the extreme condition changes the order of magnitude of the metrics. In the high-frequency test $Y_{32}^{16}$, since the gradient is larger than in the low-frequency test $Y_2^2$, the impact is

correspondingly larger. The metric $l_2$ reaches approximately $5 \times 10^{-2}$ in $Y_{32}^{16}$ using the extreme representative. For the low-frequency test $Y_2^2$, the metric $l_2$ is amplified to reach approximately $6 \times 10^{-3}$.

Metric $l_\infty$, a measure of local influence, shows more impact than the $l_2$ metric. In the high-frequency test $Y_{32}^{16}$, the extreme representative longitude amplifies the local discrepancies by roughly $2 \times 10^{-1}$ or more. An insignificant impact is shown in the above tests for small deviations of the representative longitude (Fig. 3c). The extreme representative longitude amplifies the

impact by approximately $1 \times 10^{-2}$ or more.

Figures 4 and 5 show the results of remapping using three choices of the representative longitude for test $Y_2^2$ and $Y_{32}^{16}$, respectively. The destination grid cells are set as $(N_\theta, N_\phi) = (512, 1024)$. Figure 4(b) is the remapping result for the case of $\alpha = 0$ that is expected to provide the correct solution in the RLL rectangle case. Figure 4(d) corresponds to the result of the case of $\alpha = 0.5$, which has the maximum deviation from the pivot longitude within a source cell. In these plots, as well as in

Figs. 5(b) and (d), differences relative to the exact field (a) are hardly visible. Figure 4(f) corresponds to the result of the case of $\phi_{\mathsf{rep}} = 180°$, which is an extreme deviation from the pivot longitude. Jagged patterns are visible as the distance from the $\phi_{\mathsf{rep}}$ increases.

Figure 4(c) shows the difference between the result for $\alpha = 0$ and the exact field (b−a). A cyclic pattern is apparent where the maximum and minimum points correspond to those with zero gradient. Figure 4(e) shows the difference between the results

for $\alpha = 0.5$ and $\alpha = 0.0$ (d−b). The impact of the maximum deviation of the representative longitude from the pivot within a source cell is minor compared to the errors shown in (c). The cyclic pattern is still apparent since the deviation is common among the cells. Figure 4(g) shows the difference between the result for $\phi_{\mathsf{rep}} = 180°$ and $\alpha = 0.0$ (f−b). Here, the impact of the deviation is significant. The influence on the longitude around $\phi_{\mathsf{rep}} = 180°$ is zero since, for the particular cells along the meridian, the representative longitude matches the pivot longitude. The influence increases as the distance from the $\phi_{\mathsf{rep}}$

---

[2]There is a limit in the SCRIP-p implementation in this impractical case: the longitude relative to the representative must be within $-180°$ to $180°$; to work correctly, this is the only choice. For normal applications, this causes no problem.

 

**Table 1.** Summary of results of the sensitivity experiments for extreme representative longitude $\phi_{\text{rep}} = 180°$, using Schemes O, P, Cd, and Cg. The metric $l_2$ is shown. The second column indicates the representative longitude, where mid and ext correspond to the $\alpha = 0$ and $\phi_{\text{rep}} = 180°$ cases, respectively.

(a) $Y_2^2$, $l_2$

| Scheme | Rep. | $(128, 256)$ | $(256, 512)$ | $(512, 1024)$ |
|---|---|---|---|---|
| O | mid | $1.45797 \times 10^{-4}$ | $1.70181 \times 10^{-4}$ | $1.77052 \times 10^{-4}$ |
| O | ext | $5.45804 \times 10^{-3}$ | $6.10280 \times 10^{-3}$ | $6.25367 \times 10^{-3}$ |
| P | mid | $1.45797 \times 10^{-4}$ | $1.70181 \times 10^{-4}$ | $1.77052 \times 10^{-4}$ |
| P | ext | $1.45797 \times 10^{-4}$ | $1.70181 \times 10^{-4}$ | $1.77052 \times 10^{-4}$ |
| Cd | mid | $1.59711 \times 10^{-4}$ | $1.88688 \times 10^{-4}$ | $1.96646 \times 10^{-4}$ |
| Cd | ext | $1.59711 \times 10^{-4}$ | $1.88688 \times 10^{-4}$ | $1.96646 \times 10^{-4}$ |
| Cg | mid | $1.59707 \times 10^{-4}$ | $1.88684 \times 10^{-4}$ | $1.96642 \times 10^{-4}$ |
| Cg | ext | $1.59707 \times 10^{-4}$ | $1.88684 \times 10^{-4}$ | $1.96642 \times 10^{-4}$ |

(b) $Y_{32}^{16}$, $l_2$

| Scheme | Rep. | $(128, 256)$ | $(256, 512)$ | $(512, 1024)$ |
|---|---|---|---|---|
| O | mid | $4.45001 \times 10^{-3}$ | $5.39849 \times 10^{-3}$ | $5.65725 \times 10^{-3}$ |
| O | ext | $5.00119 \times 10^{-2}$ | $5.59493 \times 10^{-2}$ | $5.73419 \times 10^{-2}$ |
| P | mid | $4.45001 \times 10^{-3}$ | $5.39849 \times 10^{-3}$ | $5.65725 \times 10^{-3}$ |
| P | ext | $4.45001 \times 10^{-3}$ | $5.39849 \times 10^{-3}$ | $5.65725 \times 10^{-3}$ |
| Cd | mid | $4.44910 \times 10^{-3}$ | $5.39731 \times 10^{-3}$ | $5.65601 \times 10^{-3}$ |
| Cd | ext | $4.44910 \times 10^{-3}$ | $5.39731 \times 10^{-3}$ | $5.65601 \times 10^{-3}$ |
| Cg | mid | $4.44962 \times 10^{-3}$ | $5.39792 \times 10^{-3}$ | $5.65664 \times 10^{-3}$ |
| Cg | ext | $4.44962 \times 10^{-3}$ | $5.39792 \times 10^{-3}$ | $5.65664 \times 10^{-3}$ |

increases. Qualitatively similar results are obtained for the case of high-frequency input field $Y_{32}^{16}$ shown in Fig. 5. In particular, Fig. 5(e) already shows a slightly larger impact than that shown in Fig. 4(e).

     The fundamental problem proposed in the present study relates only to the second-order remapping weights in the longitudinal direction. Thus, the impact of the inconsistent formulation on the centroid/pivot depends on the characteristics of the input gradient field. As shown above, the larger the gradient, the more impact on the metrics. The two experiments described here

are merely for demonstration. Given that there are many other cases of input field characteristics to remap, it is quite possible that the fundamental problem can have a much greater impact than that shown in these demonstration cases. Depending on the accuracy required in a particular application, the results may suffer significantly from the inconsistent formulation. Moreover, the representative longitudes specified by the user may be inconsistent with the pivot longitude by some amount. Especially in







(a) $\overline{\Psi}_{\text{exact}}$, $Y_2^2$

(b) $\overline{\Psi}_{\text{num}}[\alpha = 0]$

(c) $\overline{\Psi}_{\text{num}}[\alpha = 0] - \overline{\Psi}_{\text{exact}}$

(d) $\overline{\Psi}_{\text{num}}[\alpha = 0.5]$

(e) $\overline{\Psi}_{\text{num}}[\alpha = 0.5] - \overline{\Psi}_{\text{num}}[\alpha = 0]$

(f) $\overline{\Psi}_{\text{num}}[\phi_{\text{rep}} = 180°]$

(g) $\overline{\Psi}_{\text{num}}[\phi_{\text{rep}} = 180°] - \overline{\Psi}_{\text{num}}[\alpha = 0]$

**Figure 4.** Results of sensitivity experiments to the representative longitude discrepancies. The test input field is $Y_2^2$. (a) An 'exact' solution obtained by Eq. (48). (b) Remapped field using Scheme O with $\alpha = 0$. (c) Difference between (b) and (a). (d) Remapped field using Scheme O with $\alpha = 0.5$. (e) Difference between (d) and (b). (f) Remapped field using Scheme O with $\phi_{\text{rep}} = 180°$. (g) Difference between (f) and (b). Resolution of the destination grid cells is $(N_\theta, N_\phi) = (512, 1024)$.





**Figure 5.** The same as Fig. 4, with test field of $Y_{32}^{16}$.





**Table 2.** The same as Tab. 1, but the metric $l_\infty$ is shown.

(a) $Y_2^2, l_\infty$

| Scheme | Rep. | $(128, 256)$ | $(256, 512)$ | $(512, 1024)$ |
|---|---|---|---|---|
| O | mid | $2.11535 \times 10^{-4}$ | $3.50802 \times 10^{-4}$ | $4.63570 \times 10^{-4}$ |
| O | ext | $1.00028 \times 10^{-2}$ | $1.50540 \times 10^{-2}$ | $1.75934 \times 10^{-2}$ |
| P | mid | $2.11535 \times 10^{-4}$ | $3.50802 \times 10^{-4}$ | $4.63570 \times 10^{-4}$ |
| P | ext | $2.11535 \times 10^{-4}$ | $3.50802 \times 10^{-4}$ | $4.63570 \times 10^{-4}$ |
| Cd | mid | $2.11669 \times 10^{-4}$ | $3.51111 \times 10^{-4}$ | $4.63969 \times 10^{-4}$ |
| Cd | ext | $2.11669 \times 10^{-4}$ | $3.51111 \times 10^{-4}$ | $4.63969 \times 10^{-4}$ |
| Cg | mid | $2.11588 \times 10^{-4}$ | $3.51112 \times 10^{-4}$ | $4.63969 \times 10^{-4}$ |
| Cg | ext | $2.11588 \times 10^{-4}$ | $3.51112 \times 10^{-4}$ | $4.63969 \times 10^{-4}$ |

(b) $Y_{32}^{16}, l_\infty$

| Scheme | Rep. | $(128, 256)$ | $(256, 512)$ | $(512, 1024)$ |
|---|---|---|---|---|
| O | mid | $1.07466 \times 10^{-2}$ | $1.88986 \times 10^{-2}$ | $2.49168 \times 10^{-2}$ |
| O | ext | $1.97823 \times 10^{-1}$ | $2.96753 \times 10^{-1}$ | $3.47658 \times 10^{-1}$ |
| P | mid | $1.07466 \times 10^{-2}$ | $1.88986 \times 10^{-2}$ | $2.49168 \times 10^{-2}$ |
| P | ext | $1.07466 \times 10^{-2}$ | $1.88986 \times 10^{-2}$ | $2.49168 \times 10^{-2}$ |
| Cd | mid | $1.04579 \times 10^{-2}$ | $1.82571 \times 10^{-2}$ | $2.40471 \times 10^{-2}$ |
| Cd | ext | $1.04579 \times 10^{-2}$ | $1.82571 \times 10^{-2}$ | $2.40471 \times 10^{-2}$ |
| Cg | mid | $1.04624 \times 10^{-2}$ | $1.82638 \times 10^{-2}$ | $2.40549 \times 10^{-2}$ |
| Cg | ext | $1.04624 \times 10^{-2}$ | $1.82638 \times 10^{-2}$ | $2.40549 \times 10^{-2}$ |

cases involving irregular shapes of the source grid cells, this may substantially reduce the accuracy of the field after remapping.
Because of this, it is recommended that the user review all results using the J99 algorithm to confirm the level of accuracy.

### 3.3 Performance of a sample implementation

The performance of the three proposals offered as alternatives to the inconsistent J99 formulation of the second-order conservative remapping weights is shown here through an illustrative example. The results of the sensitivity experiment described in the previous section are shown first in order to confirm the insensitivity of the formulation to the representative longitude
specification. Figure 3 shows the metrics $l_2$ and $l_\infty$ with Scheme P as the dotted line. As clearly shown, Scheme P is not affected by the choice of representative longitude. This was similarly confirmed for the other two schemes, Schemes Cg and Cd (not shown). Tables 1 and 2 summarize the extreme representative longitude case using the three schemes, which also shows no difference between the $\alpha = 0$ and $\phi_{\text{rep}} = 180°$ cases. As described above, Scheme P is compatible formulation to Scheme O when $\alpha = 0$, which is confirmed by these results.



In terms of the metrics, the differences in the results of Scheme Cg/Cd and Scheme Cd are minor compared to those of
Scheme P, indicating that the flux approximation is basically the same for Schemes Cg and Cd. The implementation of Scheme
Cd involves replacing the formulation of the input flux gradient, while Scheme Cg is easy to introduce into the program, with
comparable remapping results.

While the metric values using Scheme P are less than those using Schemes Cg and Cd for the low-frequency test $Y_2^2$, they are
slightly higher for the high-frequency test $Y_{32}^{16}$ in the experiment in the present study. Which is better for the general problem is
difficult to conclude. The primary difference in Scheme P and Scheme Cg or Cd is the formulation of the flux approximation;
thus, one approximation may be better for one field, while the other is better for another field. More sensitivity experiments
involving the flux in practical cases are necessary to advance the discussion; however, this is left to future studies. In the present
paper, multiple choices are offered.

Figures 6 and 7 show the results of remapping using the three proposed schemes for test fields of $Y_2^2$ and $Y_{32}^{16}$, respectively.
As shown in lefthand panels of the two figures, differences in the remapped fields are again hardly visible.

Figures. 6(b) and 7(b) show the differences between Scheme P and Scheme O for input fields $Y_2^2$ and $Y_{32}^{16}$, respectively. Here,
Scheme O corresponds to the experiments in the previous section, with $\alpha = 0$. As expected, identical results are confirmed.

Figures. 6(d) and 7(d) show the difference between the results of Scheme P and Scheme Cd. The minor difference relative
to the input field has already been shown, reflecting the difference in the formulation of the flux approximation. Figures. 6(f)
and 7(f) show the difference between the Scheme Cd and Scheme Cg results. Here, an even smaller difference than that in (d)
is indicated. This difference reflects the treatment of the cosine term in the computation of the remapping weights. In Scheme
Cd, the flux approximation follows Eq. (38), which is provided outside the algorithm, and the flux derivatives are evaluated at
the midpoint of the source cell. In Scheme Cg, the same flux divided by cosine mid-latitude (Eq. 41) is given as an input. The
input flux gradient is adjusted by the centroid latitude (Eq. 42), not by the mid-latitude, to compute the remapping weights.
Therefore, even though the remapping is based on the same flux derivative, a small deviation occurs in the remapping results.

### 3.4    Additional remarks — applications in past studies

Two reports relating to the second-order conservative remapping scheme based on J99 are worth noting.

Ullrich et al. (2009) present a remapping scheme called Geometrically Exact Conservative Remapping (GECoRe) and show
its performance in idealized cases, comparing it with other schemes, including SCRIP. They find that the error measures in
GECoRe and SCRIP deviate significantly for the second-order methods, where the former produces results one or two orders
of magnitude better than the latter.

Mahadevan et al. (2022) present another intercomparison study using four remapping algorithms with distinct design ap-
proaches. One of the algorithms is the Earth System Modeling Framework (ESMF) regridder (Hill et al., 2004), which imple-
ments a first- and second-order conservative remapping scheme based on J99. In their results, even though the second-order
method has better accuracy than the first-order method, the authors show degraded convergence rates (a metric used for the
intercomparison; see the paper for details) compared to those expected. It is also shown that, among the four algorithms fea-
tured, the ESMF algorithms were more accurate for one remapping type, which was called RLL-CS type in Mahadevan et al.





**Figure 6.** Results of remapping using the three alternate schemes proposed in the study. The test input field is $Y_2^2$. (a) Remapped field using Scheme P. (b) Difference between (a) and Scheme O (Fig. 4b). (c) Remapped field using Scheme Cg. (d) Difference between Scheme Cg and Scheme P. (e) Remapped field using Scheme Cd. (f) Difference between Scheme Cd and Scheme P.




(a) $\overline{\Psi}_{\mathrm{num}}[\mathrm{P}]$        (b) $\overline{\Psi}_{\mathrm{num}}[\mathrm{P}] - \overline{\Psi}_{\mathrm{num}}[\mathrm{O}]$

(c) $\overline{\Psi}_{\mathrm{num}}[\mathrm{Cd}]$        (d) $\overline{\Psi}_{\mathrm{num}}[\mathrm{Cd}] - \overline{\Psi}_{\mathrm{num}}[\mathrm{P}]$

(e) $\overline{\Psi}_{\mathrm{num}}[\mathrm{Cg}]$        (f) $\overline{\Psi}_{\mathrm{num}}[\mathrm{Cg}] - \overline{\Psi}_{\mathrm{num}}[\mathrm{Cd}]$

**Figure 7.** The same as Fig. 6, with test field of $Y_{32}^{16}$.



(2022). (As noted earlier, RLL stands for regular latitude-longitude; CS stands for cubed-sphere.) The author speculates that the computation of remapping weights for RLL-CS types is based on the former mesh (RLL), in which the pivot longitude is identical to the midpoint of the longitude range. Another remapping type in their tests is MPAS-RLL, where MPAS indicates quasi-uniform Voronoi. The reason that ESMF has less accuracy for the MPAS-RLL remapping type than the RLL-CS type, even though both involve RLL, might be that the remapping weights computation in MPAS-RLL is not on the RLL but on the MPAS grid cells.

Although this may not fully explain the behaviors in the above studies, it is consistent with the prospect described in the present paper that an irregular source grid shows inferior results to those for the RLL rectangle source grid because of the mid-longitude side effect.

Valcke et al. (2022) present yet another intercomparison study using four remapping algorithms, including SCRIP. A few results obtained by SCRIP are analyzed in the paper, which shows no significant deviation from the other three algorithms, at least for the second-order conservative remapping. The result plots show that the misfit by SCRIP is the largest among the four algorithms for one benchmark, while it is far less for the other benchmark. Since, in principle, the effect of representative/pivot discrepancies is unexpected, a more detailed exploration of the source code and data is required in order to determine precisely what is happening.

## 4   Summary and conclusion

In this paper, the second-order conservative remapping method on spherical coordinates proposed by J99 is reformulated in an effort to remove the inconsistencies discovered in the original formulation. Three proposals are presented for the valid formulation of the source flux approximation and centroid (or pivot) constraints used to compute the remapping weights. The resulting weights were confirmed to be insensitive to the choice of longitude origin. Until now, the native implementation package of the original algorithm SCRIP has served to mask the inconsistency in the original formulation, as an adjustment to the relative longitude in the SCRIP code has tended to minimize or even erase the problem. Given the adjustment in SCRIP, the author believes that in most practical cases, those using the second-order remapping algorithm in J99 will experience no significant negative impact from the inconsistency problem, especially for cases involving RLL rectangular grid cells. However, it may be prudent for those conducting studies that involve irregularly shaped grid cells or non-modest variable fields and require a high degree of accuracy to review relevant prior studies.

The present study is by no means meant to denigrate past research. To the contrary, the author truly appreciates the contributions of past studies and the accompanying programming packages, which have played an invaluable role in the efforts of the entire climate modeling community. This paper is not intended to discourage but rather to support their continued use. If this were not the case, the author would have sought only to develop a new programming package without suggesting revisions to the native SCRIP package. SCRIP-p, a fork of SCRIP, can serve as a drop-in replacement for the original version, acting as a bridge until an official package revision. It should be recognized, however, that SCRIP-p was examined on a somewhat limited





basis and for only a few cases. Although it may not fully resolve the fundamental problem for general cases, it is hoped that it will work well as a first trial.

*Code and data availability.* The official package of SCRIP version 1.5 is available from github: https://github.com/SCRIP-Project/SCRIP (last access: 1 April 2024), under an open-source license, with copyright owned by the Regents of the University of California. Details of
500 the license are described in a document of the package. SCRIP-p, a fork of SCRIP, is available from github: https://github.com/saitofuyuki/ scrip-p (last access: 1 April 2024), with the same license as the official package, except for where modified, whose copyright is owned by Japan Agency for Marine-Earth Science and Technology (JAMSTEC) under Apache license version 2.0. The exact version of the official and the fork packages, as well as input data and scripts, used to produce the results used in this paper are archived on Zenodo (https: //doi.org/10.5281/zenodo.10892796).

**Appendix A:  Centroid versus Pivot**

To distinguish the physical spherical coordinate system from the logical plane coordinate system, additional notation is introduced as needed. The physical spherical coordinates are represented as $(\hat{\theta}, \hat{\phi}, \hat{\rho})$, while the logical plane coordinates are $[\theta, \phi]$. The centroids in the two systems are called the physical centroid and the logical centroid, respectively.

In principle, the integral in Eq. (7) can be evaluated under the physical spherical coordinate system. The flux approximation
Eq. (2) is reformulated using the physical coordinate vectors:

$$f_n = f(\hat{\mathbf{c}}_n) + \nabla_n f \cdot (\hat{\mathbf{r}} - \hat{\mathbf{c}}_n), \tag{A1}$$

where the vectors and the gradient term are expressed using a three-dimensional Euclidean basis. The position vector $\hat{\mathbf{r}} = [x, y, z]^\mathsf{T}$ of the Cartesian coordinate is expressed using the spherical coordinate components $(\hat{\theta}, \hat{\phi}, \hat{\rho})$ as

$$\hat{\mathbf{r}} = \left[ \hat{\rho} \cos \hat{\theta} \cos \hat{\phi}, \ \hat{\rho} \cos \hat{\theta} \sin \hat{\phi}, \ \hat{\rho} \sin \hat{\theta} \right]^\mathsf{T}, \tag{A2}$$

which can be inverted to $(\hat{\theta}, \hat{\phi}, \hat{\rho})$ as

$$\begin{cases} \hat{\rho} = \sqrt{x^2 + y^2 + z^2}, \\ \hat{\theta} = \arcsin(z/\hat{\rho}), \\ \hat{\phi} = \arctan(y/x). \end{cases} \tag{A3}$$

The centroid $\hat{\mathbf{c}}_n$ is represented similarly using components $(\hat{\theta}_c, \hat{\phi}_c, \hat{\rho}_c)$. A straightforward way to compute the physical centroid is to first transform to a three-dimensional Cartesian vector representation, since latitude and longitude are not suitable for this objective (Jenness, 2011). Computing the surface integral for each Cartesian component transforms back to the spherical
coordinate component of latitude, longitude, and radius. The physical centroid $\hat{\mathbf{c}}_n$ defined on the three-dimensional Euclidean space computed by this method has no relationship to the logical centroid $\mathbf{c}_n$ using Eqs. (19) and (20). (Indeed, the author had





attempted to implement the second-order conservative remapping method in this this way but failed to achieve the expected accuracy.) The differences and similarities can be demonstrated using a simple grid configuration.

To illustrate, an RLL rectangular grid cell bounded by two parallels, $\theta = \theta_0$, $\theta = \theta_1$, and two meridians, $\phi = \phi_0$, $\phi = \phi_1$, is adopted. Introducing Cartesian coordinates of the centroid as $\hat{\mathbf{c}}_n = [c_x, c_y, c_z]^{\mathsf{T}}$, these coordinates are computed, followed by conversion to the spherical coordinates. The $x$ component of the centroid of an RLL rectangular cell can be computed as follows:

$$
\begin{aligned}
c_{x\,\mathsf{RLL}} &= \int_{\phi_0}^{\phi_1}\int_{\theta_0}^{\theta_1} \left[\cos^2\hat\theta\cos\hat\phi\right]\mathrm{d}\hat\phi\,\mathrm{d}\hat\theta \bigg/ \int_{\phi_0}^{\phi_1}\int_{\theta_0}^{\theta_1} \left[\cos\hat\theta\right]\mathrm{d}\hat\phi\,\mathrm{d}\hat\theta \\
&= \frac{\sin\left(\dfrac{\phi_1+\phi_0}{2}\right)\cos\left(\dfrac{\phi_1-\phi_0}{2}\right)\left[\sin(2\hat\theta)+2\hat\theta\right]_{\theta_0}^{\theta_1}}{2(\phi_1-\phi_0)(\sin\theta_1-\sin\theta_0)},
\end{aligned}
\tag{A4}
$$

and, similarly, for the other two components as

$$
c_{y\,\mathsf{RLL}} = \frac{\sin\left(\dfrac{\phi_1+\phi_0}{2}\right)\sin\left(\dfrac{\phi_1-\phi_0}{2}\right)\left[\sin(2\hat\theta)+2\hat\theta\right]_{\theta_0}^{\theta_1}}{2(\phi_1-\phi_0)(\sin\theta_1-\sin\theta_0)},
\tag{A5}
$$

$$
c_{z\,\mathsf{RLL}} = \frac{\sin\theta_1+\sin\theta_0}{2},
\tag{A6}
$$

using trigonometric identities. Finally, they are transformed back to spherical coordinates:

$$
\hat\rho_{c\,\mathsf{RLL}} = \sqrt{(c_{x\,\mathsf{RLL}})^2 + (c_{y\,\mathsf{RLL}})^2 + (c_{z\,\mathsf{RLL}})^2},
\tag{A7}
$$

$$
\hat\theta_{c\,\mathsf{RLL}} = \arcsin\left(c_{z\,\mathsf{RLL}}/\hat\rho_{c\,\mathsf{RLL}}\right),
\tag{A8}
$$

$$
\hat\phi_{c\,\mathsf{RLL}} = \arctan\left(c_{y\,\mathsf{RLL}}/c_{x\,\mathsf{RLL}}\right) = \frac{\theta_0+\theta_1}{2}.
\tag{A9}
$$

No simple formulation may be derived except for the centroid longitude. For RLL grid cells, the centroid longitude matches the midpoint of the longitude range. As naturally expected, if the shape of a source grid cell is symmetric along a meridian, the longitudes always match the meridian in order to balance.

The pivot latitude on the logical plane of the RLL rectangle can be derived using Eq. (17) as follows:

$$
\begin{aligned}
\theta_{p\,\mathsf{RLL}} &= \int_{\phi_0}^{\phi_1}\int_{\theta_0}^{\theta_1} \left[\theta\cos\theta\right]\mathrm{d}\phi\,\mathrm{d}\theta \bigg/ \int_{\phi_0}^{\phi_1}\int_{\theta_0}^{\theta_1} \left[\cos\theta\right]\mathrm{d}\phi\,\mathrm{d}\theta \\
&= \frac{\left[\theta\sin\theta+\cos\theta\right]_{\theta_0}^{\theta_1}}{\sin\theta_1-\sin\theta_0},
\end{aligned}
\tag{A10}
$$




where the area element $dA = \cos\theta\,d\theta\,d\phi$ is applied. The pivot longitude of the RLL rectangle can be derived as

$$\phi_{p\mathsf{RLL}} = \int_{\phi_0}^{\phi_1}\int_{\theta_0}^{\theta_1}\left[\phi\cos^2\theta\right]d\phi\,d\theta \bigg/ \int_{\phi_0}^{\phi_1}\int_{\theta_0}^{\theta_1}\left[\cos^2\theta\right]d\phi\,d\theta \tag{A11}$$

$$= \frac{\phi_0 + \phi_1}{2}.$$

Similarly, the logical centroid can be formulated in this way. From Eq. (19), $\theta_{c\mathsf{RLL}} = \theta_{p\mathsf{RLL}}$, and the longitude of the logical centroid is computed as

$$\phi_{c\mathsf{RLL}} = \int_{\phi_0}^{\phi_1}\int_{\theta_0}^{\theta_1}\left[\phi\cos\theta\right]d\phi\,d\theta \bigg/ \int_{\phi_0}^{\phi_1}\int_{\theta_0}^{\theta_1}\left[\cos\theta\right]d\phi\,d\theta \tag{A12}$$

$$= \frac{\phi_0 + \phi_1}{2},$$

which is also the same as $\phi_{p\mathsf{RLL}}$.

The longitudes of two types of centroids and the pivot all match the middle point of the RLL rectangle cell. On the other hand, the latitudes of the physical centroid and the other two are expected to be different quantities, since the former depends on the cell longitudes $\phi_0$ and $\phi_1$ through $\rho_c$ (Eq. A8), whereas the latter two do not (as shown in Eq. A10). Therefore, in principle, the centroid can never be an alternate of the pivot in spherical coordinates.

Obviously, the centroid is not necessarily on the sphere surface. In the context of a remapping application in climate modeling, however, such extreme situations are rare. It is likely that a grid cell is small enough to assume that the radial component is very close to the unit ($\rho \approx 1$). Under this assumption, the centroid latitude can be approximated as

$$\theta_{c\mathsf{RLL}} \approx \arcsin(c_{z\mathsf{RLL}}) = \arcsin\left(\frac{\sin\theta_1 + \sin\theta_0}{2}\right). \tag{A13}$$

This approximated latitude corresponds to the latitude that divides the source RLL cell into two equal areas, which is analogous to the case of a centroid of a plane rectangle. The approximation is now independent from the cell longitudes; however, the latitude formulation Eq. (A13) is still different from the pivot, Eq. (A10).

Figure A1 shows a comparison of the pivot and centroid latitude for RLL rectangular grid cell cases. The latitude grid lines (cell corner coordinates) are expressed as $\theta = 180°(j - N_\theta/2), j = 0, \cdots, N_\theta$. Uniform spacing of the longitude cells as $\Delta\phi = 360°/N_\phi$ is adopted. Four cases of $(N_\theta, N_\phi)$ are examined: $(64, 128)$, $(128, 256)$, $(256, 512)$, and $(512, 1024)$.

The results show that the physical centroid latitude (solid lines) computed by Eq. (A8) is constantly biased toward the pole from the logical pivot latitude (they are antisymmetric along the equator). The bias is largest for the mid-latitude.

On the other hand, the centroid latitude approximated by Eq. (A13) is biased toward the equator (thus, a negative difference is shown in the figure). The magnitude of the bias increases toward the pole. It is common that the bias becomes smaller as the resolution increases, but the approximated centroid latitude shows a relatively large deviation from the pivot at the pole.

The lower figure shows the normalized difference between the two latitudes. $|(\sin\hat{\theta}_c - \sin\theta_p)/(\sin\theta_1 - \sin\theta_0)|$ is shown for all the grid cells. The approximated centroid shows a normalized difference of approximately $5\%$ for the cell at the highest



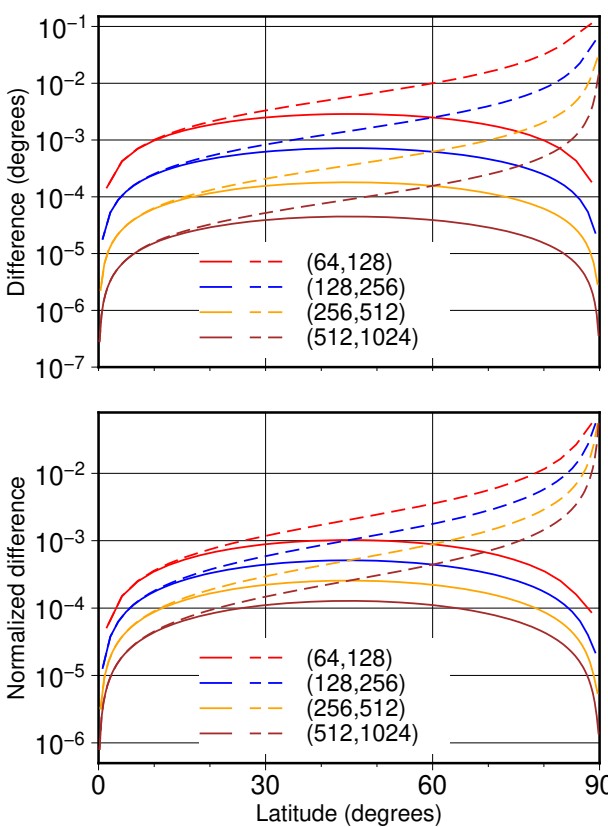

**Figure A1.** Comparison of the pivot and physical centroid latitudes in the case of an RLL rectangular cell. The cases of total cell numbers $(N_\theta, N_\phi) = (64, 128)$, $(128, 256)$, $(256, 512)$, and $(512, 1024)$ are displayed with equally spaced latitude coordinates. Differences in the computed latitudes relative to the logical pivot latitude, Eq. (A10), are shown. The solid lines correspond to the physical centroid latitude, Eq. (A8), while the dashed lines are the approximated centroid latitude, Eq. (A13), with negation of the sign (the difference is always negative). The upper and lower figures show the difference and normalized difference, i.e., the difference between sine latitudes normalized by the cell sine latitude range, respectively. The figure is symmetric along the equator, thus only a positive latitude area is shown.



latitudes for all resolutions, while the physical latitude shows, at most, a $0.1\%$ deviation. Thus, the physical centroid coordinates computed on the Euclidean space do not significantly deviate from those on the logical plane, at least for RLL rectangular cells.

*Author contributions.* FS did all the work in the present paper.

*Competing interests.* The author declares that they have no conflicts of interest



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
