# Peer review of "Centroids in second-order conservative remapping schemes on spherical coordinates"

_EGUsphere, 2024_

## Referee Comment (RC4)

**1. OVERALL REVIEW**

The manuscript is well-written, the derivations are clear, and the arguments are coherent. However, some major issues need to be addressed to verify and demonstrate the properties of the new remapping scheme variants introduced here.

The fundamental mismatch in the derivation presented here occurs due to an incorrect transformation from a spherical coordinate system to a cylindrical projection onto a 2D logical plane. Since the discussions are primarily restricted to RLL grids on a logical plane, these coordinate transformations play a role in computing the actual weights. The author assumes that the position vector $r = [\theta, \phi] = \theta e_\theta + \phi e_\phi$ instead of the J99 assumption of using $r = [\theta, \phi] = \theta e_\theta + cos(\theta)\phi e_\phi$. The author should address these concerns and verify if the conclusions differ.

The author presents variants of the second-order scheme. However, discussions related to accuracy for the choice of representative coordinates and its impact on accuracy measures should be accompanied by a convergence order study. It is important to understand and verify the rate of convergence, and the constant involved to see if the new schemes offer a significantly better advantage in terms of stability and accuracy for remapping fields conservatively. This is mandatory for a comparative study presented here.

I also recommend using the MIRA package (referenced in Mahadevan et al (2022)) to generate the metrics data for remapping a given analytical field (both spherical low/high order harmonics functionals and a double vortex field) to understand stability, conservation, and accuracy degradations if any in $L_2, L_\infty, H_1$ norms. Such a study can provide better intuition on the numerical performance and asymptotic behavior of the remapping method.

**2. NOTABLE COMMENTS**

Other major comments are listed below.

1. What is the relevance of Eq (8)? This is the same as Eq (2) except that Eq (4) has been substituted in. This discussion can be simplified.

2. L96: "The author speculates that it is non-trivial to satisfy transformation from Eq. (5) to Eq. (6) for general coordinates.". If you use a consistent linear basis for the reconstruction with a constant gradient across a cell, then this should be true. What do you mean by "general coordinates" here?

3. Eq (9) is true for a rectangular projection of a spherical coordinate system defined on the surface of a unit sphere. Please be explicit about this if you claim it "is just the analogue to the (x,y) Cartesian representation".

4. Is it correct that $\sigma$ density term in Eq (12) refers to the physical coordinate transformation on the unit spherical surface to the logical lat-lon 2D plane coordinate system? There is no further discussion related to this term, which I think is necessary to set up the derivations that follow.

5. In Eq (13), the second term in the integral equals zero according to the assumption in Eq (14). However, even with the assumption that the flux derivatives are constant across a cell, I fail to see how the individual terms are equated to zero in Eq (15) and Eq (16). Is this imposed specifically to derive what the optimal pivot coordinates need to be? This is only a sufficient condition and not a necessary condition.

6. I do not see a clear reason why $cos(\theta_p)$ was substituted with $cos(\theta)$ in Eq (11). You replaced a point

value with a spatially varying term, which leads to differences in Eq (18) and Eq (20). This seems to be a key argument stating that the centroid and the pivot on the logical plane are different in longitudinal direction. However, if you had retained $cos(\theta_p)$, the formulations will be identical. This is also mentioned in L151.

- Edit: After reading Phil's review comments, the reasoning is clearer.

7.  Eq (31) implies that J99 is using $A_i = \int_i dA_{s,i} = \int_i cos(\theta)dA$, where $A_i$ is the area of the logical element $i$, and $A_{s,i}$ is the area of a spherical element $i$. With the definition of $dA = cos(\theta)d\theta d\phi$, the derivation of $w_{3nk}$ looks consistent. This negates the conclusion that J99 derivation yields a wrong remapping weight term. Please clarify as this is one of the primary conclusions that drives the motivation for the manuscript.

8.  Scheme $C_g$ seems like an approximation of Scheme P, where $cos(\theta)$ is replaced by $cos(theta_c)$ everywhere and simplified. In that respect, it is closer to Scheme P than Scheme $C_d$ in contrary to what the author has suggested in L315.

9.  Is the $\phi_{rep}$ defined in Eq (49) used to replace the center latitude-longitude values in the input grid file so that the reference J99 implementation uses it as is without modifications? It is unclear in the text and I see `src_grid_center_lat` and `src_grid_centroid_lat` in the testO/rmp map files distributed in the artifact at DOI:10.5281/zenodo.10892795. Please clarify.

10.  In Fig (3), can you explain the smaller differences in $l_2$ metric between $Y_2^2$ and $Y^1{}_63 2$ as compared to $l_\infty$, which indicates a contrasting behavior? Can you also comment on whether the larger errors near the poles are dominating in these metrics? This may be important since it is my understanding that there is a separate treatment for elements at the poles compared to everywhere else.

11.  In Fig (4), why are figures 4(d) and 4(f) compared against 4(b), instead of 4(a). You have established in Table (1) that Scheme O (J99) is sensitive to $\alpha$. So error differences against the exact solution will provide a better way to compare profiles in Fig 4(c) against 4(e) and 4(g).

The same comment applies to Fig (5) as well.

12.  I recommend replacing Fig (5) with a similar experiment as Fig (4) using Scheme P instead of Scheme O.

13.  L435: "Which is better for the general problem is difficult to conclude."

Certainly. But since the manuscript is focused on the consistency of second-order schemes, you should use the analytical closed for functionals to compute the order of convergence going from say a refined RLL grid (1024,2048) to (90,180), (180,360), (360, 720), (720, 1440). The source and destination grids mustn't be embedded to avoid any aliasing errors to creep in. Such a convergence study can also provide insight into the constant in the second-order scheme that will determine overall accuracy measures.

14.  Another suggestion here is to use the dual-stationary vortex (Nair and Machenhauer, 2002) as another test case to verify the performance of the schemes.

15.  Fig (6) and Fig (7): It is unclear which scheme is better or what the real conclusions are from these results. What do the changes in Schemes $C_g$ and $C_d$ relative to scheme P tell you? There is no clear value in this particular experiment and the text does not explain the significance of this result either. Please clarify, and improve the text/figures appropriately.

**3.  MINOR COMMENTS**

1.  L49: Add "grid": regular latitude-longitude (RLL) rectangle grid.

2. L64: Add comma, after "in a conservative manner"

3. Eq (27) and Eq (28): please stay consistent with notation; use J99 instead of ORG

4. Eq (29): Do not change bracket notation unless you intend to specify something different. For example, $(\theta - \theta_p)$ in Eq (25) is replaced by $[\theta - \theta_p]$ in Eq (29).

5. L218: please specify that $\phi_{rep}$ is the representative coordinate, eventhough this is mentioned again later

6. L359: "using the official SCRIP implementation."

7. L426: Rephrase: "This was similarly confirmed for the other two schemes, Schemes Cg and Cd (not shown)."

8. L430: "in the results of Scheme Cg/Cd and Scheme Cd" - remove the first **/Cd** mention?

**REFERENCES**

Nair, R. D., and B. Machenhauer, 2002: The mass-conservative cell-integrated semi-Lagrangian advection scheme on the sphere. Mon. Wea. Rev., 130, 649–667, doi:10.1175/1520-0493(2002)

---

## Author Comment (AC2)

**Response to the Reviewer comments (RC1)**

I thank to the reviewer Dr. Moritz Hanke who provided precise and valuable feedbacks on the manuscript. I addressed all the points in the responses as follows, and I will submit the revised manuscript that reflects these changes, which significantly improves the quality of the manuscript.

The reviewer comments are quoted in italic with some minor editorial adjustments, followed by responses by the author.

*Fuyuki Saito found an error in the formulation of the weights for second order conservative remapping paper by P. W. Jones from 1999 (referred to as J99). This error also made it into the SCRIP library, which is based on J99. This software library has been the basis for conservative interpolation in many climate models for many years, which makes the finding presented here a substantive contribution to the community. Unfortunately my limited understanding of the underlying math prevents me from commenting on the correctness of the presented formulas.*
*The author reproduces the results from J99 and describes in detail the error made therein. This is followed by an analysis of the SCRIP source code and the description of preprocessing step in SCRIP. This step avoids the error having an actual impact on the interpolation results for many common use-cases. Multiple solutions to problem are derived and explained in-depth. Afterwards the impact of error is analyzed in great detail and compared to presented solutions.*
*Overall the paper is very well written and after a minor revisit, I would recommend it for publishing.*

Thank you very much for your summary and the positive evaluation.

*General remarks:*
*Even through J99 and the SCRIP library have been widely used in the past, the author seems to wrongfully assume that this is still the case for current software (e.g. ESMF). This is apparent in the additional remarks and the summary of the paper. However, ESMF and other software (e.g. YAC or XIOS) use implementations for first and second order conservative remapping, which significantly differ from the SCRIP library and are therefore not prone to the presented error. Second order conservative remapping in these implementations is based on Kritsikis et al., 2017 and not J99.*

I really appreciate that you have kindly instructed me the recent situation. I fully agree that J99 and the SCRIP library were used in the past, and the communities have been switching to the other software as you tell. However, as far as I surveyed, some (although not many) recent studies still use the J99 scheme for the second-order conservative method. For example, Ding et al. (2024), Chtirkova et al. (2024) and Ren and Zhou (2024) explicitly mention that they used the second-order conservative remapping scheme of J99.

*In the summary of the paper the author encourages the further use of the SCRIP library. However, he fails to mention the various other drawbacks of it, which would lead me to a different conclusion. These drawbacks include inaccuracy for cells close to the poles (also mention by the author) due to how trigonometric functions are used for intersection computation and the misrepresentation of the true cell shapes for everything but RLL rectangular grids (Taylor, 2024). This limits the use of the SCRIP library in my opinion to remapping between two RLL rectangular grids, which could be implemented much simpler and more accurate.*

Yes, I agree that I may have encouraged too much in the manuscript. I do not suggest to replace the other remapping softwares by the SCRIP, but just want to support the past and current application of J99 and SCRIP.

Therefore, in order to respond the two comments above, in the revised manuscript I will summarize the current situation around J99 and SCRIP: First, there are other softwares with different implementation

from J99 or SCRIP and their application is recently increased significantly. Except for the RLL rectangular grids, the J99 algorithm has drawbacks which may reduce the remapping accuracy. The other softwares are not based on J99 algorithm, and the argument of the present paper do not affect their application. Second: there a few recent studies still utilize the J99 or SCRIP to compute the second-order conservative remapping. Even the argument of the present paper (i.e., the derivation of J99 is invalid) is correct, it may not mean that the result and interpretation is completely wrong.

***Specific comments:***
***Line 3: missing reference for "1987" and "1999"***

The abstract will be much rewritten after restructuring of the manuscript, and this part will be no more exist.

***Line 3-4: duplicated "has"***

I will removed the first one.

***Line 5: missing reference for "pioneer study"***

Also this part will be fully rewritten.

***Line 18: Conservative interpolation is just one of multiple methods being used in ESM's. However, this sentence implies that it is the most commonly used method.***

I agree. Conservation properties is merely one choice for the users. Some spatial interpolation schemes without conservation are also used in common. I will change this sentence to mention about this aspect.

***Line 34-37: The CDO's use code extracted from YAC for the first order conservative interpolation. (see Taylor, 2024)***

Right. I will clarify this point, where only the second-order scheme in CDO is based on SCRIP.

***Line 42: Give Equation number from J99 ("(10)"?).***

I will insert the equation number.

***Section "Introduction": This sections could mention other issues of SCRIP and solutions to this implemented in other software (see General remarks).***

Right. It is good place to include the other SCRIP issues. Other referees have told the issues in their review, which will be included here in the revised manuscript.

***Line 73 Eq. (2): "r" is not described***

I will define $r$ as the point vector.

***Figure 2: Instead of showing the actual source code, a mathematical description of what the code does might be easier to understand.***

It sounds better. I will introduce the mathematical description corresponding to the actual implementation, possibly with the source code moved to Appendix.

***Line 245: This may not only happen at the poles. A cell with the longitude bounds of [179;181] may be represented by [179;-179] independent of that latitude.***

Right. I will revise the original sentence to describe this is an example.a

**Line 205: The title could be more concise. In general this paragraph contains in my view too much speculations and opinions of the author. It could probably be shortened without loosing significant information.**

How about 'A relative longitude formulation in the native package'? Although it is not shorter, I suppose it be more concise to describe the subsection.

**Line 256: The implementations of trigonometric function can be more accurate for small absolute values. This may be another reason for this code in SCRIP.**

I agree it is another merit of relative coordinate. Actually other referee (Jones) has pointed out that the intention of this adjustment is to simplify multiple-valued coordinate in the longitude, with citing the original J99 paper. I will correct the sentence according to the original intention.

**Line 366, 375, 381: repeated use of "naturally...within the cell"**

I agree, they are verbose. I will clean these parts.

**Section "Additional remarks": In this section the discussion of the results for ESMF and other software tested by Valcke, 2022 assumes that it is based on J99, which is not the case.**

I suppose the above point is not by Valcke(2022) but Mahadevan (2022). Thanks a lot for this information. I am not aware that the ESMF in Mahadevan (2022) is not based on J99. I will rewrite or remove this part.

**Line 380: "less impact than a change in magnitude" this is not further explained and quantified. Is this due to numerical inaccuracy of SCRIP for cells close to the poles?**

This part is really confusing. What I am describing here is the difference between the results of O and P (solid and dashed line in Fig 3) is very small compare to the magnitude of scores. I will rewrite this part to be clearer.

**Line 399: "which has the maximum deviation from the pivot longitude within a source cell" this has already been explained above**

I will delete the repetition.

**Line 403: Maybe you could explicitly mention that Figure 4(c) shows deviations from the exact fields, which are independent from the issue discussed in this paper. Additionally, you could state that in order to be able to visualize this issue, you have to compute the difference (d-b/f-b) instead of (d-a/f-a).**

Yes. I will introduce a description to mention the Fig 4(c) does not directly related to the topic of the paper but just a demonstration. The difference plotted in the figures are already d-b (Fig. 4e), f-b (Fig. 4g) as you suggested. I will rewrite the captions to be clearer.

**Line 522: duplicated "this"**

I will delete it.

**Line 578,585,594: duplication of "https://doi.org"**

Thanks a lot. I will check all the references in the revised manuscript.

**References**

Chtirkova, B., Folini, D., Ferreira Correa, L., and Wild, M.: Shortwave Radiative Flux Variability Through the Lens of the Pacific Decadal Oscillation, Journal of Geophysical Research: Atmospheres, 129, e2023JD040 520, https://doi.org/10.1029/2023JD040520, e2023JD040520 2023JD040520, 2024.

Ding, S., Zhi, X., Lyu, Y., Ji, Y., and Guo, W.: Deep Learning for Daily 2-m Temperature Downscaling, Earth and Space Science, 11, e2023EA003 227, https://doi.org/10.1029/2023EA003227, e2023EA003227 2023EA003227, 2024.

Ren, Z. and Zhou, T.: Understanding the alleviation of "Double-ITCZ" bias in CMIP6 models from the perspective of atmospheric energy balance, Climate Dynamics, https://doi.org/10.1007/s00382-024-07238-7, 2024.

---

## Author Comment (AC3)

**Response to the Reviewer comments (RC2 and RC3)**

I thank to the reviewer Dr. Phil Jones who provided precise and valuable feedbacks on the manuscript. In particular, I really appreciate that you provide the detail information about the original formulation that I missed to catch in the manuscript. I addressed all the points in the responses as follows, and I will submit the revised manuscript that reflects these changes, which significantly improves the quality of the manuscript.

The reviewer comments are quoted in italic with some minor editorial adjustments, followed by responses by the author.

**1 General comments on RC2 and RC3**

At beginning, I want to clarify the situation of the present paper. The referee commented in RC3 as follows:

*Ah, yes. Did get a bit sloppy/inconsistent there. I probably should have stuck with the position vector here (phi only in the centroid) and included the cos(theta) metric only when computing distance as in the flux expansion.*

Thus it is now agreed that there is an inconsistency in the formulation of the original paper (Jones, 1999, hereafter referred to as J99). Indeed, that is exactly the inconsistency I argued in the present paper.

The essential point of the inconsistency comes from the series of formulation in the original paper J99:

$$f_n = \overline{f}_n + \nabla_n f \cdot (\mathbf{r} - \mathbf{r}_n), \tag{J99.5}$$

$$\mathbf{r}_n = \frac{1}{A_n} \int_{A_n} \mathbf{r} \, dA, \tag{J99.6}$$

$$\overline{F}_k = \sum_{n=1}^{N} \left[ \overline{f}_n w_{1\,nk} + \left( \frac{\partial f}{\partial \theta} \right)_n w_{2\,nk} + \left( \frac{1}{\cos \theta} \frac{\partial f}{\partial \phi} \right)_n w_{3\,nk} \right], \tag{J99.7}$$

$$w_{3\,nk} = \frac{1}{A_k} \int_{A_{nk}} \cos\theta(\phi - \phi_n)\, \mathrm{d}A \qquad \text{(J99.10)}$$

$$= \frac{1}{A_k} \int_{A_{nk}} \phi\cos\theta\, \mathrm{d}A - \frac{w_{1\,nk}}{A_n} \int_{A_n} \phi\cos\theta\, \mathrm{d}A,$$

where Eq. (J99.5) corresponds to Eq. (5) in J99 and so on. All the above essential equations originate from J99, thus it does not matter to the main point of the present paper, what formulations are inserted before or appended after these combination, even if they do not match the intention in the original paper. The derivation using 2D logical plane in the manuscript is misleading as you get much confused. I am really sorry about that and it is completely my fault in the first manuscript. However, it is only a preparation to obtain Eq. (J99.7).

The final formulation of the centroid longitude that you presented in RC3 is as follows:

$$\cos\theta\phi_n = \frac{\int_{A_n} \phi\cos\theta\, \mathrm{d}A}{\int_{A_n} \mathrm{d}A}, \qquad \text{(RC2.7)}$$

while my proposal in the manuscript is:

$$\phi_p = \left( \int_{A_n} \phi\cos\theta\, \mathrm{d}A \right) \Big/ \left( \int_{A_n} \cos\theta\, \mathrm{d}A \right). \qquad \text{(S24.18)}$$

In the manuscript, in order to avoid the confusion of the formulation, I call the reference coordinate as 'pivot' and use $\phi_p$ symbol. Whatever I call it, my suggestion is to replace the original formulation of the centroid longitude by Eq. (S24.18).

These two equation differ only in the treatment of $\cos\theta$ term in an integral. If $\cos\theta$ term in the integral of the denominator of Eq. (S24.18) is extracted from the integral as it is, the equation would become identical with Eq. (RC2.7).

You agree with the inconsistency I proposed and suspect it insignificant, as commented in RC3 (to follow the above quotation):

*In the end, I suspect it may not make a large difference - basically the difference between average of the product and product of the average. But better to be consistent.*

This is true, however, only when the computation of the centroid longitude is done using the longitude coordinate relative to the centroid. Yes, this is a recursion: the centroid longitude is computed with the centroid longitude. Actually, it is much close to what the original algorithm is computing — instead of the centroid, the source cell center is adopted for the reference of relative longitude. The center position of cell is usually not far from the centroid, and thus the effect of inconsistency is kept sufficiently small.

In order to evaluate the influence of the inconsistency, here I show a simple demonstration. Actually, this corresponds to how I found the consistency. For the demonstration, please forget about the formulation of Cartesian coordinate. The following speculation just starts from the latter half of Eq. (J99.10) on the spherical coordinate.

$$w_{3\,nk} = \frac{1}{A_k} \int_{A_{nk}} \phi \cos\theta \; \mathrm{d}A - \frac{w_{1\,nk}}{A_n} \int_{A_n} \phi \cos\theta \; \mathrm{d}A. \tag{1}$$

Introducing $w_{1\,nk}$, it is reformulated as follows:

$$A_k \cdot w_{3\,nk} = \int_{A_{nk}} \phi \cos\theta \; \mathrm{d}A - \frac{A_{nk}}{A_n} \int_{A_n} \phi \cos\theta \; \mathrm{d}A. \tag{2}$$

If the cell is a RLL shape, the integral can be computed as follows:

$$\int_{\mathrm{RLL}} \phi \cos\theta \; \mathrm{d}A = \int_{\phi_0}^{\phi_1} \int_{\theta_0}^{\theta_1} \phi \cos^2\theta \; \mathrm{d}\phi \, \mathrm{d}\theta \tag{3}$$

$$= \left[ \frac{\phi_1^2 - \phi_0^2}{2} \right] \left[ \frac{(\sin 2\theta_1 + 2\theta_1) - (\sin 2\theta_0 + 2\theta_0)}{4} \right] \tag{4}$$

$$= \left[ \frac{\phi_1^2 - \phi_0^2}{2} \right] \mathsf{S}(\theta_0, \theta_1), \tag{5}$$

where the coordinates of the corners of the RLL grid are $\phi_0$, $\phi_1$, $\theta_0$, $\theta_1$, and $\mathsf{S}(\theta_0, \theta_1)$ is introduced to represent the second bracket term.

For example, we can compute the weight in a very simple case (see Fig. 1): a RLL source cell is divided into two equal-area RLL along the latitude. The source cell is assumed to be $\phi_0$, $\phi_1$, $\theta_0$, $\theta_1$ RLL shape. Then the cell $A_{nk}$ is $\phi_0$, $\phi_1$, $\theta_0$, $\theta_c$ RLL cell, where the corner latitude $\theta_c$ is defined as:

$$\sin\theta_c = \frac{\sin\theta_0 + \sin\theta_1}{2}. \tag{6}$$

[Figure]

Figure 1: Example configuration

Actually, since the longitude span of cell $A_{nk}$ is the same as the source cell, the weight in the longitudinal direction becomes zero (for the J99 algorithm). Then Eq. (2) is computed as follows:

$$A_k \cdot w_{3\,nk} = \left[\frac{\phi_1^2 - \phi_0^2}{2}\right] \left[\mathsf{S}(\theta_0, \theta_c) - \frac{A_{nk}}{A_n}\mathsf{S}(\theta_0, \theta_1)\right] \tag{7}$$

$$= \left[\frac{\phi_1^2 - \phi_0^2}{2}\right] \left[\mathsf{S}(\theta_0, \theta_c) - \frac{1}{2}\mathsf{S}(\theta_0, \theta_1)\right], \tag{8}$$

where $A_{nk}/A_n = 1/2$ by definition, and it is expected to be 0. The second bracket term is not zero even with substituting $\theta_c$ term as Eq. (6). Therefore, it means that Eq. (7) is satisfied only when $\phi_1 = \phi_0$ (trivial, zero width), and when $\phi_0 = -\phi_1$. The latter relation means that the origin is the center point of the source cell. Thus, if we compute each weight with rotating the longitude such that the origin correspond to each cell center, then the weight is correctly computed as zero. If the longitudes are not specified relative to the cell center, then it suffers from the absolute value of the longitude, which can be significantly large at the worst case. This example is only a simple RLL case, but it is sufficient to show that the original formulation of the weight breaks the expected behavior.

Please remember again that the above speculation does not depend on the incorrect assumption of plane coordinate in my manuscript.

This inconsistency originates from the treatment of $\cos\theta$ term in Eq. (RC2.7). If you keep the $\cos\theta$ term inside the integral as Eq. (S24.18), then the complex $\mathsf{S}$ representation in Eq. (8) is simplified and, actually, it is equivalent to zero for any absolute longitude values. Thus the weights can be computed as expected wherever the longitude origin is located.

Also, the speculation above is on a simple RLL case as a demonstration. Generally, the treatment of $\cos\theta$ term in the original algorithm does not satisfy the expected characteristics coming from the flux assumption of Eqs. (J99.5) and (J99.7) in J99 paper, which again are independent on the incorrect assumption of my manuscript.

So, I will reconstruct the discussion using the spherical coordinate as a starting point throughout the revised manuscript. I suppose then I can explain my derviation much clearer than the first one. I will withdraw the story of Taylor series expansion from the revised manuscript. I hope you would be satisfied with my proposal, after restructuring of my derivation from the beginning.

**2    Point-to-point comments on RC2**

*I will first note in this review that my original publication (J99) and subsequent implementation is far from perfect and has some serious issues. I've always wanted to go back and correct those but unfortunately never got the time to do so. I say this to emphasize that the critical review below is not meant as a reactive defense of J99 or the SCRIP implementation. However, in reviewing the paper, I found the author has made some significant errors and incorrect assumptions that negate the conclusion. I do not believe this paper can be published in its present form since it is incorrect.*

First of all, I really appreciate you kindly to become a referee of this paper. Your comment will really fill the gap of my understanding the J99 and SCRIP. I may agree that this paper cannot be published in its present form, not because it is incorrect but it starts from different assumption to the J99.

Even I starts from those you provided below, I still suppose it is transformed into the invalid formulation as speculated above.

*The first error is the derivation in section 2.1. The author attempts to derive the flux distribution from a Taylor series expansion. However the constraints in equations (4),(5) do not necessarily follow from (3) or at least not uniquely so. They are merely a reasonable and obvious choice among a number of possible solutions. For this reason, neither Dukowicz and Kodis (DK87) nor J99 derive this form from a Taylor series. The two previous papers (JK87, J99) simply show that the flux form:*

$$f_n = \overline{f}_n + \nabla_n f \cdot (\mathbf{r} - \mathbf{r}_n), \qquad \text{(RC2.1)}$$

*meets the conservation condition as long as the reference point ($\mathbf{r}_n$) used in the flux approximation is the centroid. It is an assumed distribution that meets the conservation condition. This might seem a minor quibble since the author arrives in a similar place as the two prior papers, but it is important because the author takes the Taylor series approach later as well and this is incorrect.*

Response to the series of paragraphs from here will be inserted at the end of the series.

*The author correctly notes that equation (J99.6) only holds for Cartesian coordinates. In spherical coordinates, the dot product and the unit vectors are spatially dependent and cannot be formally pulled out of the integral and are more complex in form. The original paper J99 arrives at this form in a different manner. There is a similar issue with the centroid definition, but I will address both of these choices below.*

*The real problems with the current paper come in section 2.2 where the author incorrectly represents the J99 derivation by stating we use a Cartesian space in lat/lon. This is not the case. All of our derivation occurs in spherical coordinates or a local spherical surface*

*approximation with the appropriate metric factors included. The author can be excused in misunderstanding the derivation since much of the derivation is left to the reader in the original J99 paper. But this mistaken assumption leads to the incorrect form that is the core of the paper.*

*To elucidate the error, I have to explain how we actually derive the weights in J99 and show some additional steps. We start with the form of the flux approximation shown above (noting again, that this is not a Taylor series expansion):*

$$f_n = \overline{f}_n + \nabla_n f \cdot (\mathbf{r} - \mathbf{r}_n). \qquad \text{(RC2.1 revisited)}$$

*We also assume the gradient is fixed with the form:*

$$\nabla f_n = \left(\frac{\partial f}{\partial \theta}\right)_n \hat{\theta} + \left(\frac{1}{\cos\theta}\frac{\partial f}{\partial \phi}\right)_n \hat{\phi}. \qquad \text{(RC2.2)}$$

*As noted previously the dot product in spherical coordinates is in general not simply a component-wise product as in Cartesian coordinates since the unit vectors can change direction based on position on the sphere. Here we can take two approaches which lead to the same approximation. One is to say that the unit vectors are nearly aligned so that local orthogonality is almost true (it is true for r in this case, but not quite $\mathbf{r}_n$). We use the fact that the local displacement on the unit sphere is:*

$$d\mathbf{r} = d\theta\hat{\theta} + r\cos\theta\,d\phi\hat{\phi}. \qquad \text{(RC2.3)}$$

*Then we can approximate the flux as:*

$$f_n = \overline{f} + \left(\frac{\partial f}{\partial \theta}\right)_n (\theta - \theta_n) + \left(\frac{1}{\cos\theta}\frac{\partial f}{\partial \phi}\right)_n \cos\theta(\phi - \phi_n). \qquad \text{(RC2.4)}$$

*The same result can be obtained by using a local quasi-Cartesian approach but including the spherical metric factors. This form is very close to the author's equation (11) except that the first term is the mean flux and the author's pivot point cannot be an arbitrary pivot point. It is required to be the centroid $\mathbf{r}_n$. These differences again*

*arise from the mistaken use of a Taylor expansion rather than the flux form.*

*At this point, the author makes the mistake at the core of the paper. The author assumed we were working in some sort of Cartesian space in (theta,phi) with none of the metric factors and then assumes that a density factor is required in equation (12) to correct the integrals and must occur in all integrals. In fact, we are working in spherical coordinates where the area element dA is defined as*

$$\mathrm{d}A = \cos\theta \, \mathrm{d}\theta \, \mathrm{d}\phi. \tag{RC2.5}$$

*So we do not need this imagined sigma density in equation (12) and equation (18) is incorrect in the denominator.*

*In J99, we compute the centroid, using the standard definition*

$$\mathbf{r}_n = \frac{\int_{A_n} \mathbf{r} \, \mathrm{d}A}{\int_{A_n} \mathrm{d}A}. \tag{RC2.6}$$

*This leads to the correct latitude centroid in equation (17). The position vector r must be dimensionally (and metrically) consistent with the displacement equation above, so the centroid term in longitude includes the cos(lat) metric scale factor and*

$$\cos\theta\phi_n = \frac{\int_{A_n} \phi\cos\theta \, \mathrm{d}A}{\int_{A_n} \mathrm{d}A}. \tag{RC2.7}$$

*Substituting this form of the centroid, we obtain equation (28) (equation 10 in J99). The author's equation 30,31 are incorrect since they depend on the incorrect equation (12). As another aside, we note that a more careful computation of a real centroid should follow, for example, the approach Du, Gunburger and Ju (2003, SIAM J. Sci Computing) in which the centroid is the full 3d centroid constrained to the spherical surface. The centroid here is a very close approximation to that form and is consistent with the dot product assumptions made in the flux approximation so that actual conservation is ensured in practice.*

Thanks a lot for the detail explanation. Now it is really clear to me how to formulate the original equation. I am really happy to learn the background of J99, which definitely must replace my wrong assumption in the manuscript.

I fully agree that to start from a Taylor series expansion is one of possible choices, and does not match to the derivation of the original paper. However, as you may agree, even if I start from your derivation, the final formulation for the weight in the longitudinal direction, with regard to the treatment of the cosine latitude term.

***The author's error in equation 12 renders the remaining discussion essentially moot. However, I will note that in the later discussion the author misunderstands the longitudinal correction shown in Figure 2. This correction is necessary to avoid computing a line integral across the branch cut in the multiple-valued longitude as explained in section 3(d)(1) of J99.***

Again, I am really sorry to confuse you by the wrong assumption of the starting point of the original formulation. I will reconstruct the discussion and reformulate everything using the spherical coordinate as a starting point throughout the paper, and I hope then you proceed to the core formulation of my derivation in the revised manuscript.

Also, thanks a lot for this point, about the longitudinal correction. I should have noticed this point when I read J99 many times repeatedly. So now I understand what is the actual intention to introduce relative longitude to compute the remapping weights.

Even if the objective of the longitudinal correction is for simply treatment of multiple-valued longitude, it still works as a side-effect to reduce the inconsistency in the original formulation. There is no strong constrain to adopt the source cell 'center' for the reference point of longitude correction. For this particular objective, any longitude within the source cell (for example, one of the source cell boundary) should work as the same. However, as I demonstrated in the manuscript, remapping is a little but certainly influenced by the choice of the reference point.

What I examined in the manuscript is to shift `grid_center_lon` variable

within source cells in the SCRIP input file and input to the original SCRIP. The remapped fields are affected slightly but significantly.

I admit that the demonstration of this impact is not clear enough (similar concerns are raised in RC4, by Mahadevan). I will reconstruct the discussion and make it clearer in the revised manuscript.

*I hope this review is clear and helps the author understand what was done for J99 and the SCRIP implementation. As noted at the beginning, there are other problems with SCRIP, especially the parametric form for cell sides used in equations 16, 17,18 of J99 and in the SCRIP implementation. Both the choice of a linear form and the failure to include the cos(lat) metric factors in the longitude during intersection computations (even though I did so everywhere else in the paper) are the source of most of the problems in SCRIP and are especially amplified in the 2nd-order terms, making SCRIP difficult to us for higher-order cases.*

Yes, it is really clear, helps enough to improve the manuscript. Also, thanks a lot for the comment on the original implementation. This should be also mentioned in the revised manuscript.

**3 Point-to-point comments on RC3**

*Ah, yes. Did get a bit sloppy/inconsistent there. I probably should have stuck with the position vector here (phi only in the centroid) and included the cos(theta) metric only when computing distance as in the flux expansion. Guess I got used to tossing in the metric factor everywhere and it does make for a cleaner weight calculation and is probably better behaved. In the end, I suspect it may not make a large difference - basically the difference between average of the product and product of the average. But better to be consistent.*

Actually, that inconsistency is the topic of my manuscript. It has been insignificant for the past application of the original algorithm, not because the difference between average of the product and product of the average is small,

but because the relative longitude formulation works to cancel the inconsistency as a side effect, which is demonstrated in the general comment. I will reformulate the discussion to be clarified my point.

**References**

Jones, P. W.: First- and Second-Order Conservative Remapping Schemes for Grids in Spherical Coordinates, Monthly Weather Review, 127, 2204 – 2210, 1999.

---

## Author Comment (AC4)

**Response to the Reviewer comments (RC4)**

I thank to the reviewer Dr. Vijay Mahadevan who provided precise and valuable feedbacks on the manuscript. I addressed all the points in the responses as follows, and I will submit the revised manuscript that reflects these changes, which significantly improves the quality of the manuscript.

The reviewer comments are quoted in italic with some minor editorial adjustments, followed by responses by the author.

*1. OVERALL REVIEW The manuscript is well-written, the derivations are clear, and the arguments are coherent. However, some major issues need to be addressed to verify and demonstrate the properties of the new remapping scheme variants introduced here.*

Thanks a lot for, in particular, positive evaluation on the derivation in the present paper. Although my derivation in the manuscript is not yet accepted by everybody, I am really encouraged by your review. I suppose I can manage to respond all the criticism on the derivation, with including your suggestion to be added into the revised manuscript.

*The fundamental mismatch in the derivation presented here occurs due to an incorrect transformation from a spherical coordinate system to a cylindrical projection onto a 2D logical plane. Since the discussions are primarily restricted to RLL grids on a logical plane, these coordinate transformations play a role in computing the actual weights. The author assumes that the position vector $r = [\theta, \phi] = \theta e_\theta + \phi e_\phi$ instead of the J99 assumption of using $r = [\theta, \phi] = \theta e_\theta + \cos(\theta)\phi e_\phi$. The author should address these concerns and verify if the conclusions differ.*

The main point of the present paper is about following three equations, (6) (7) (10) in the J99 paper:

$$\mathbf{r}_n = \frac{1}{A_n} \int_{A_n} \mathbf{r} \, dA, \qquad\qquad (J99.6)$$

$$\overline{F}_k = \sum_{n=1}^{N} \left[ \overline{f}_n w_{1\,nk} + \left(\frac{\partial f}{\partial \theta}\right)_n w_{2\,nk} + \left(\frac{1}{\cos\theta}\frac{\partial f}{\partial \phi}\right)_n w_{3\,nk} \right], \qquad (J99.7)$$

$$w_{3\,nk} = \frac{1}{A_k} \int_{A_{nk}} \cos\theta (\phi - \phi_n)\, \mathrm{d}A \qquad\qquad \text{(J99.10)}$$

$$= \frac{1}{A_k} \int_{A_{nk}} \phi \cos\theta \ \mathrm{d}A - \frac{w_{1\,nk}}{A_n} \int_{A_n} \phi \cos\theta \ \mathrm{d}A.$$

What I proposed is that those three equations are inconsistent. The derivation using 2D logical plane in the manuscript is somewhat misleading, but, it is only a preparation to obtain Eq. (J99.7), and has little influence on the conclusion. Thanks to RC2, RC3 (referee comments by reviewer 2, Jones), I can formulate Eq. (J99.7) starting from the spherical coordinate. In the revised manuscript, I will rewrite the preparation block with the 2D logical plane into the one with the spherical coordinate. It will be much clearer to satisfy your concern. Thanks a lot.

*The author presents variants of the second-order scheme. However, discussions related to accuracy for the choice of representative coordinates and its impact on accuracy measures should be accompanied by a convergence order study. It is important to understand and verify the rate of convergence, and the constant involved to see if the new schemes offer a significantly better advantage in terms of stability and accuracy for remapping fields conservatively. This is mandatory for a comparative study presented here.*

First of all, the primary issue of the present paper is (was) not to propose a new scheme, but to demonstrate the possible impact of the error (still just a proposal, not accepted by everyone) in the past application.

I wrote in the manuscript that the choice of representative coordinate is in principal not under control of the algorithm, but as pointed in RC2 (Jones), it is recommended already in J99 to choose the source cell center in order to simply deal with multiple-valued longitudes. It means that for most practical cases, the representative coordinates are always close to the centroid coordinates, which will result in insignificant influence on the remapped field. Thus the impact on accuracy measures should be almost identical with a fully compatible implementation with the original. The extreme representative

coordinate experiment will be reduced or even removed in the revised paper, because it may be far from practical application.

In this perspective, the suggestion above may be too much for the revised manuscript, because it is beyond the scope of the main topic, to show that the past application using J99 algorithm is little damaged by the original formulation.

However, the present manuscript may be regard as one to propose some variants of the second-order scheme, as commented. They are all minor variations, not so different from the original algorithm, but still it may be useful to present a convergence order study.

Therefore I am now much postive to include your suggestion. Thanks a lot for this comment. It will improve the quality of the revised manuscript.

Also, as I wrote in the manuscript, computation of the remapping weights using the relative coordinates still works as a side effect to reduce the error of invalid derivation. The error is canceled only in the case when the representative (source cell center) coincides with the centroid of the source cell. Although cell center usage is a simple way, it is possible to overcome the multiple-valued longitude in different ways. The three variations of second-order scheme are insensitive to the choice of the representative coordinate while the original is much sensitive. In this sense, a convergence order study for cases of extreme representative coordinate might be still useful. I will consider whether or not include after restructuring the main part of the present paper.

*I also recommend using the MIRA package (referenced in Mahadevan et al (2022)) to generate the metrics data for remapping a given analytical field (both spherical low/high order harmonics functionals and a double vortex field) to understand stability, conservation, and accuracy degradations if any in $L_2, L_\infty, H_1$ norms. Such a study can provide better intuition on the numerical performance and asymptotic behavior of the remapping method.*

Great. I agree to introduce MIRA for the metrics.

**2. NOTABLE COMMENTS Other major comments are listed below.**

**1. What is the relevance of Eq (8)? This is the same as Eq (2) except that Eq (4) has been substituted in. This discussion can be simplified.**

Right. I will delete the equation (8) to be simplified, such as "Under Eq. (7) constraints with Eq. (4), the flux approximation (Eq. 2) automatically satisfies the conservation characteristics of Eq. (1)."

**2. L96: "The author speculates that it is non-trivial to satisfy transformation from Eq. (5) to Eq. (6) for general coordinates.". If you use a consistent linear basis for the reconstruction with a constant gradient across a cell, then this should be true. What do you mean by "general coordinates" here?**

What I wanted to say is that it is sufficient if this transformation is valid for a linear basis at least for the topic of the present paper. However, I agree it is really confusing. I will delete this block to simplify.

**3. Eq (9) is true for a rectangular projection of a spherical coordinate system defined on the surface of a unit sphere. Please be explicit about this if you claim it "is just the analogue to the (x,y) Cartesian representation".**

The derivation from here will be completely rewritten according to RC2, RC3 (by reviewer 2 Jones). The core derivation is formulated using the basis of spherical coordinate.

This part will be introduced in later part, and I will be explicit about the rectangular projection at a new place.

**4. Is it correct that $\sigma$ density term in Eq (12) refers to the physical coordinate transformation on the unit spherical surface to the logical lat-lon 2D plane coordinate system? There is no further discussion related to this term, which I think is necessary to set up the derivations that follow.**

The density comes from the conservation of the flux, and simply computed from $\mathrm{d}A = \cos\theta\,\mathrm{d}\theta\,\mathrm{d}\phi = \sigma\,\mathrm{d}\theta\,\mathrm{d}\phi$. But, again, this part will be reformulated with the basis of spherical coordinate.

**5. In Eq (13), the second term in the integral equals zero according to the assumption in Eq (14). However, even with the assumption that the flux derivatives are constant across a cell, I fail to see how the individual terms are equated to zero in Eq (15) and Eq (16). Is this imposed specifically to derive what the optimal pivot coordinates need to be? This is only a sufficient condition and not a necessary condition.**

Exactly, they are imposed to derive the pivot coordinates for a source cell $A_n$, in one way among possible solution.

**6. I do not see a clear reason why $\cos(\theta_p)$ was substituted with $\cos(\theta)$ in Eq (11). You replaced a point value with a spatially varying term, which leads to differences in Eq (18) and Eq (20). This seems to be a key argument stating that the centroid and the pivot on the logical plane are different in longitudinal direction. However, if you had retained $\cos(\theta_p)$, the formulations will be identical. This is also mentioned in L151. - Edit: After reading Phil's review comments, the reasoning is clearer.**

Yes I definitely agree. It is a weak point of my original derivation, and honestly I do not have confidence to justify the replacement. In the revised manuscript, this part will be reformulated from the spherical basis, and it will be much clearer. Thanks a lot.

**7. Eq (31) implies that J99 is using $A_i = \int_i dA_{s,i} = \int_i \cos(\theta) dA$, where $A_i$ is the area of the logical element $i$, and $A_{s,i}$ is the area of a spherical element $i$. With the definition of $dA = \cos(\theta) d\theta d\phi$, the derivation of $w_{3nk}$ looks consistent. This negates the conclusion that J99 derivation yields a wrong remapping weight term. Please clarify as this is one of the primary conclusions that drives the motivation for the manuscript.**

The definition of $A_i$ you proposed above implies that the remapping coefficient $w_{1nk}$ (the first-order coefficient) should be computed also with it. The

original formulation of the integral term (J99 Eq.12) is:

$$\int_{A_{nk}} \mathrm{d}A = \oint_{C_{nk}} -\sin\theta \ \mathrm{d}\phi, \qquad (\text{J99.12})$$

is rather consistent with the other (i.e., the present paper's) definition of the area element of no additional cosine term insertion. Thanks a lot for this point. I will describe the possibility of the different formulation of area term in the revised manuscript.

**8. Scheme $C_g$ seems like an approximation of Scheme P, where $\cos(\theta)$ is replaced by $\cos(\theta_c)$ everywhere and simplified. In that respect, it is closer to Scheme P than Scheme $C_d$ in contrary to what the author has suggested in L315.**

Right. I will change the explanation following the comment above.

**9. Is the $\phi_{rep}$ defined in Eq (49) used to replace the center latitude-longitude values in the input grid file so that the reference J99 implementation uses it as is without modifications? It is unclear in the text and I see `src_grid_center_lat` and `src_grid_centroid_lat` in the testO/rmp map files distributed in the artifact at DOI:10.5281/zenodo.1089 Please clarify.**

You are right. I only modified the value of `src_grid_center_lat` and `src_grid_centroid_lat` in the SCRIP input files which is input to the original SCRIP test program. I will clarify this.

**10. In Fig (3), can you explain the smaller differences in $l_2$ metric between $Y_2^2$ and $Y_{16}^{32}$ as compared to $l_\infty$, which indicates a contrasting behavior? Can you also comment on whether the larger errors near the poles are dominating in these metrics? This may be important since it is my understanding that there is a separate treatment for elements at the poles compared to everywhere else.**

First point. I suppose that it reflects the wide insensitive area (Fig 5e) in the result of $Y_{16}^{32}$ experiment to reduce the $l_2$ metric as compared to $l_\infty$. I

will discuss further in the revised manuscript to explain these difference in detail. Second point. I should have mentioned in the text that the separate treatment for elements around the poles is switched off in this demonstration for simplicity.

**11. In Fig (4), why are figures 4(d) and 4(f) compared against 4(b), instead of 4(a). You have established in Table (1) that Scheme O (J99) is sensitive to $\alpha$. So error differences against the exact solution will provide a better way to compare profiles in Fig 4(c) against 4(e) and 4(g). The same comment applies to Fig (5) as well.**

Yes, partially. Fig 4(d) is much close to 4(b), such that difference between 4(d) and 4(a) may look similar to 4(c). On the other hand, difference between 4(f) and 4(a) will be better as you comment, I agree. I will revise this part.

Actually, as the referee 1 (Hanke) pointed out, the primary issue of the present paper is not to propose a new scheme but to present the possible impact of the error in the past application (which is expected to be reasonably small). Therefore the comments on this figure is inconsistent between referee 1 and 3. I will mix the revision according to the both two suggestion, to keep 4(e) but to replace 4(g), which will be also consistent of the primary issue of the paper.

**12. I recommend replacing Fig (5) with a similar experiment as Fig (4) using Scheme P instead of Scheme O.**

It is possible, but Scheme P is really insensitive to the choice of $\alpha$, so the figures b,d,f will be equivalent except for the minor precision. In order to confirm this, I will insert these figures in the Supplement.

**13. L435: "Which is better for the general problem is difficult to conclude." Certainly. But since the manuscript is focused on the consistency of second-order schemes, you should use the analytical closed for functionals to compute the order of convergence going from say a refined RLL grid (1024,2048) to (90,180), (180,360), (360, 720), (720, 1440). The source and destination grids mustn't be embedded to avoid any aliasing errors to creep in. Such a convergence study**

*can also provide insight into the constant in the second-order scheme that will determine overall accuracy measures.*

Honestly I felt it too much for the issue of the present paper. However, in a sense the paper may be one to propose new schemes, I agree to your suggestion, at least as additional issue. Good idea. I will revise the demonstration to overcome this issue.

*14. Another suggestion here is to use the dual-stationary vortex (Nair and Machenhauer, 2002) as another test case to verify the performance of the schemes.*

Agree. Same as above.

*15. Fig (6) and Fig (7): It is unclear which scheme is better or what the real conclusions are from these results. What do the changes in Schemes $C_g$ and $C_d$ relative to scheme P tell you? There is no clear value in this particular experiment and the text does not explain the significance of this result either. Please clarify, and improve the text/figures appropriately.*

As I described repeatedly, the primary issue of the present paper is not to propose a new scheme but to present the possible impact of the error. I will clarify the main topic through the paper. Also, when I obtained the additional experiment you have suggested, maybe I can tell which is better or not, as side information.

*3. MINOR COMMENTS 1. L49: Add "grid": regular latitude-longitude (RLL) rectangle grid.*

Clearly the word 'grid' is missing. I will correct.

*2. L64: Add comma, after "in a conservative manner"*

Comma will be inserted.

*3. Eq (27) and Eq (28): please stay consistent with notation; use J99 instead of ORG*

Good idea. ORG will be replaced by J99.

**4. Eq (29): Do not change bracket notation unless you intend to specify something different. For example, $(\theta - \theta_p)$ in Eq (25) is replaced by $[\theta - \theta_p]$ in Eq (29).**

Thanks a lot for pointing it out. I will unify the notation through the paper.

**5. L218: please specify that $\phi_{rep}$ is the representative coordinate, even though this is mentioned again later**

All right. I will insert the definition again around here.

**6. L359: "using the official SCRIP implementation."**

I will append the word 'implementation' accordingly.

**7. L426: Rephrase: "This was similarly confirmed for the other two schemes, Schemes Cg and Cd (not shown)."**

Will be rephrase as: "Similarly, Schemes Cg and Cd are not affected (not shown)."

**8. L430: "in the results of Scheme Cg/Cd and Scheme Cd" - remove the first /Cd mention?**

Right. I will delete '/Cd' in the first place.